# CCDC32 stabilizes clathrin-coated pits and drives their invagination

Ziyan Yang[1], Changsong Yang[2], Zheng Huang[1], Peiliu Xu[1], Yueping Li[1], Lu Han[1], Linyuan Peng[1], Xiangying Wei[3], John E Pak[4], Tatyana Svitkina[2], Sandra L Schmid[4,5]*, Zhiming Chen[1]*

[1]NHC Key Laboratory of Birth Defect Research and Prevention, MOE Key Laboratory of Rare Pediatric Diseases, Institute of Cytology and Genetics of School of Basic Medical Sciences & Department of Clinical Laboratory of The First Affiliated Hospital, Hengyang Medical School, University of South China, Hengyang, China; [2]Department of Biology, University of Pennsylvania, Philadelphia, United States; [3]Fuzhou Institute of Oceanography, College of Geography and Oceanography, Minjiang University, Fuzhou, China; [4]Chan Zuckerberg Biohub, San Francisco, United States; [5]Department of Cell Biology, University of Texas Southwestern Medical Center, Dallas, United States

## eLife Assessment

The manuscript presents a **valuable** finding that CCDC32, beyond its reported role in AP2 assembly, follows AP2 to the plasma membrane and regulates clathrin-coated pit assembly and dynamics. The authors further identify an alpha-helical region within CCDC32 that is essential for its interaction with AP2 and its cellular function. While live-cell and ultrastructural imaging data are **solid**, future biochemical studies will be needed to confirm the proposed CCDC32-AP2 interaction.
[Editors' note: this paper was reviewed by Review Commons.]

**Abstract** Clathrin-mediated endocytosis (CME) is essential for maintaining homeostasis in mammalian cells. Previous studies have reported more than 50 CME accessory proteins; however, the mechanism driving the invagination of clathrin-coated pits (CCPs) remains elusive. We show by quantitative live cell imaging that siRNA-mediated knockdown of CCDC32, a poorly characterized endocytic accessory protein, leads to the accumulation of unstable flat clathrin assemblies. CCDC32 interacts with the α-appendage domain (AD) of AP2 in vitro and with full-length AP2 complexes in cells. Deletion of aa78-98 in CCDC32, corresponding to a predicted α-helix, abrogates AP2 binding and CCDC32's early function in CME. Furthermore, clinically observed nonsense mutations in CCDC32, which result in C-terminal truncations that lack aa78-98, are linked to the development of cardio-facio-neuro-developmental syndrome (CFNDS). Overall, our data demonstrate the function of a novel endocytic accessory protein, CCDC32, in regulating CCP stabilization and invagination, critical early stages of CME.

***For correspondence:**
sandra.schmid@czbiohub.org
(SLS);
zhiming.chen@usc.edu.cn (ZC)

**Competing interest:** The authors declare that no competing interests exist.

## Introduction

Clathrin-mediated endocytosis (CME) regulates nutrient uptake and maintains the activity of transmembrane transporters and is thus essential for maintaining cellular homeostasis (*Kaksonen and Roux, 2018*; *Kirchhausen et al., 2014*; *McMahon and Boucrot, 2011*; *Mettlen et al., 2018*). Malfunctions of CME are strongly associated with neurological diseases, cardiovascular diseases, and cancers

(*Blue et al., 2018*; *DeMari et al., 2016*; *Elkin et al., 2015*; *Gilles Moulay et al., 2019*; *Hamdan et al., 2017*; *Manti et al., 2019*; *Sznajder and Swanson, 2019*; *Wu and Yao, 2009*). CME occurs via the assembly of clathrin triskelia into clathrin-coated pits (CCPs) that invaginate to form clathrin-coated vesicles (CCVs; *Kaksonen and Roux, 2018*; *Mettlen et al., 2018*; *Smith and Smith, 2022*). During CCP maturation, successful invagination of the clathrin coat and its underlying membrane is a key step that determines whether nascent CCPs are productive or abortive (*Aguet et al., 2013*; *Baschieri et al., 2020*; *Chen and Schmid, 2020*; *Wang et al., 2020*). Although more than 50 endocytic accessory proteins (EAPs) have been reported to be involved in the progression of CME (*Bhave et al., 2020*; *Chen and Schmid, 2020*; *Kirchhausen et al., 2014*; *McMahon and Boucrot, 2011*; *Merrifield and Kaksonen, 2014*; *Sochacki et al., 2017*; *Taylor et al., 2011*; *Traub, 2011*), the mechanism of CCP invagination remains elusive, i.e. which accessory proteins are required and how do they regulate CCP invagination?

Coiled-coil domain-containing protein 32 (CCDC32), also known as gene *C15orf57*, is a small, 185 amino acids (aa) protein that has been poorly studied and whose function was unknown. Clinical genome sequencing of three patients with cardio-facio-neuro-developmental syndrome (CFNDS) revealed three homozygous nonsense mutations that only express the first 9, 54, and 80 aa of CCDC32, respectively (*Abdalla et al., 2022*; *Harel et al., 2020*). How these loss-of-function mutations result in CFNDS remains unknown.

A genome-wide co-essential modules study (*Wainberg et al., 2021*) suggested a functional correlation between CCDC32 and AP2, whose interactions were also implied by affinity purification-mass spectroscopy (AP-MS) analysis (*Cho et al., 2022*) and co-immunoprecipitation (co-IP) assays (*Wainberg et al., 2021*). In addition, depletion of CCDC32 was observed to inhibit transferrin uptake, suggesting a role in CME (*Wainberg et al., 2021*). More recently, and while this paper was under review, *Wan et al., 2024* reported an essential role for CCDC32 as a co-chaperone with AAGAB (alpha and gamma adaptin binding protein) in the assembly of AP2 complexes. The authors showed that in CRISPR-mediated CCDC32 knockout cells, AP2 complexes were severely depleted, and CME was strongly inhibited. They further showed in vitro that CCDC32 was recruited to the AAGAB:α:σ2 hemicomplex, where it displaced AAGAB, recruited μ2 and β2 subunits to assemble the mature AP2 complex and then was released. They were unable to detect interactions between a C-terminally tagged CCDC32 and the mature AP2 complex (*Wan et al., 2024*).

Using a combination of biochemistry, quantitative live cell imaging, and ultrastructure electron microscopy, we report that a functional N-terminally tagged CCDC32 interacts with the intact AP2 complex and is recruited to CCPs. siRNA knockdown of CCDC32 inhibits CCP invagination and stabilization, without impacting the levels of AP2 expression. Thus, we demonstrate a second, critical role for CCDC32 in regulating early stages of CME by regulating the stabilization and invagination of CCPs.

## Results

### CCDC32 is recruited to CCPs

We first explored a direct role for CCDC32 in CME by testing if and when CCDC32 is recruited to CCPs. ARPE-HPV cells that stably express mRuby-clathrin light chain a (mRuby-CLCa) and siRNA-resistant, N-terminally GFP tagged full-length CCDC32 (eGFP-CCDC32(FL)) were generated (*Figure 1—figure supplement 1*) and imaged using Total Internal Reflection Fluorescence Microscopy (TIRFM). In addition to diffuse staining across the inner plasma membrane (PM) surface, suggestive of direct membrane binding, we also observed colocalization of eGFP-CCDC32(FL) and mRuby-CLCa (*Figure 1A*) at clathrin-coated pits.

Furthermore, cohort-averaged fluorescence intensity traces were obtained using time-lapse imaging and primary (mRuby-CLCa)/subordinate (eGFP-CCDC32) tracking powered by cmeAnalysis (*Aguet et al., 2013*; *Chen et al., 2020*; *Jaqaman et al., 2008*). As a negative control, ARPE-HPV cells that stably express mRuby-CLCa and eGFP showed neither diffuse PM staining nor eGFP recruitment to CCPs (*Figure 1—figure supplement 2*), whereas more than half of analyzed CCPs were observed to recruit CCDC32. The recruitment curve of CCDC32 follows the assembly of clathrin on CCPs (*Figure 1B*). Importantly, compared to CCDC32-positive CCPs, those that fail to recruit eGFP-CCDC32 exhibited significantly shorter and exponentially decaying lifetimes, previously shown to be

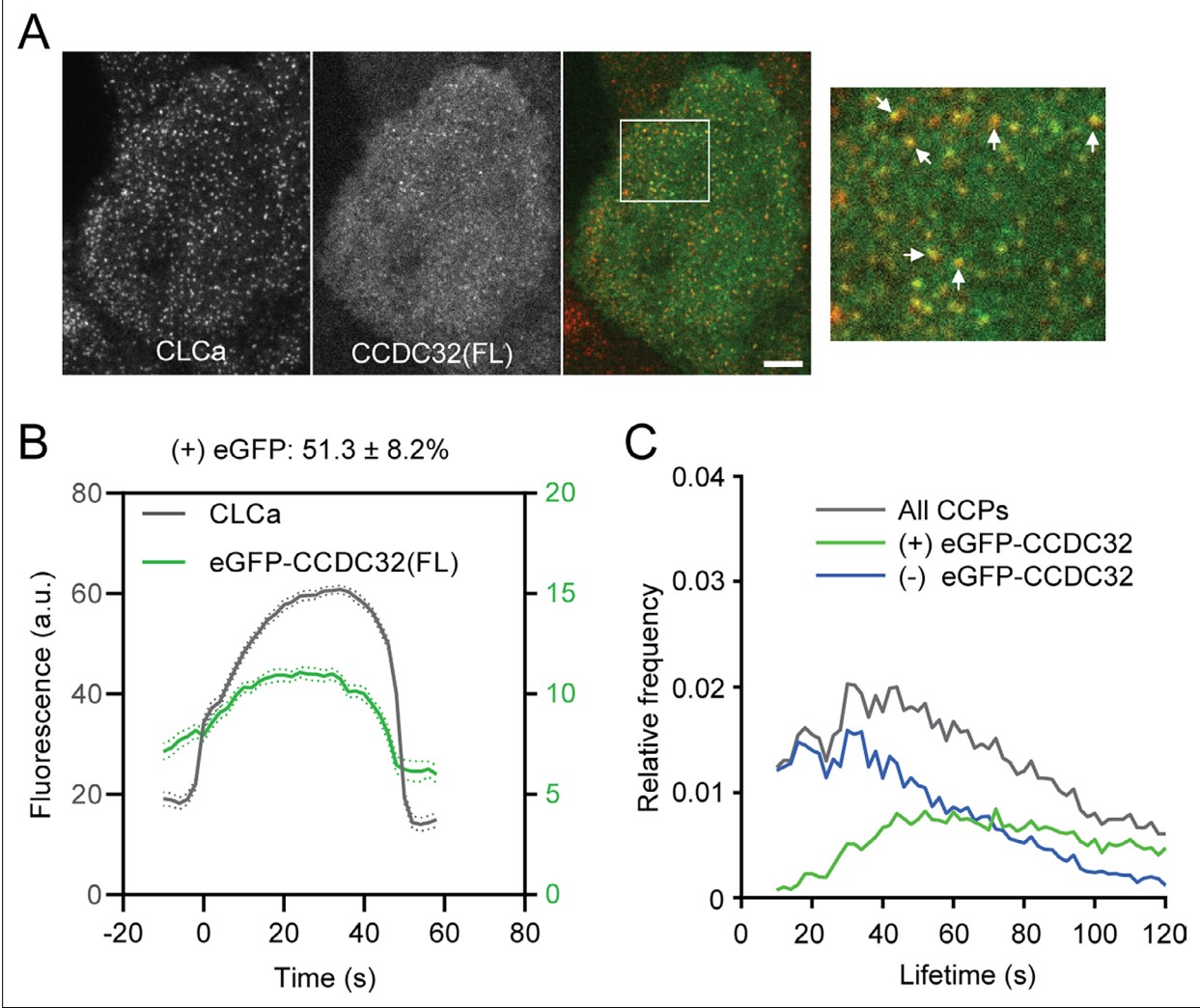

**Figure 1.** CCDC32 is recruited to clathrin-coated pits. (**A**) Representative TIRFM images of ARPE-HPV cells that stably express mRuby-CLCa and eGFP-CCDC32(FL). This dual-channel imaging was conducted without siRNA-mediated knockdown. White arrows point to colocalized CLCa and CCDC32 clusters. White ROI is magnified on the right. Scale bar = 5 μm. (**B**) Cohort-averaged fluorescence intensity traces of CCPs (marked with mRuby-CLCa) and CCP-enriched eGFP-CCDC32(FL). 51.3±8.2% of analyzed CCPs showed eGFP-CCDC32(FL) recruitment. Number of tracks analyzed: 23699. (**C**) Lifetime distributions of all CCPs, CCPs with eGFP-CCDC32 recruitment, and CCPs without eGFP-CCDC32 recruitment.

The online version of this article includes the following source data and figure supplement(s) for figure 1:

**Source data 1.** Numeric data to generate *Figure 1*.

**Figure supplement 1.** Western Blot of recombinant cells.

**Figure supplement 1—source data 1.** PDF file containing original western blots for *Figure 1—figure supplement 1*, indicating the relevant bands and treatments.

**Figure supplement 1—source data 2.** Original files for western blot analysis displayed in *Figure 1—figure supplement 1*.

**Figure supplement 2.** Dual channel TIRFM imaging revealed no eGFP recruitment to CCPs.

**Figure supplement 2—source data 1.** Numeric data to generate *Figure 1—figure supplement 2*.

characteristic of abortive CCPs (*Figure 1C*; *Aguet et al., 2013*). Together, these data establish that the early recruitment, along with clathrin, of CCDC32 to nascent CCPs regulates their maturation and lifetimes.

## CCDC32 depletion inhibits transferrin receptor (TfnR) uptake and CCP formation

To gain further insight into the function of CCDC32 in CME, we next examined the effects of siRNA-mediated knockdown of CCDC32 on CME and CCP dynamics. In agreement with the previous observation (*Wainberg et al., 2021*), siRNA-mediated knockdown of CCDC32 in ARPE-HPV cells, which resulted in an ~60% depletion of CCDC32 (*Figure 2A and B*), reduced the cellular uptake efficiency of TfnR (*Figure 2C*).

A recent study reported that CCDC32 functions as an essential chaperone for AP2 complex assembly and that CRISPR-mediated knockout of CCDC32 led to a severe decrease in AP2 (*Wan et al., 2024*). Under our conditions of CCDC32 knockdown, we did not detect any decrease in protein levels of the AP2 complex (*Figure 2—figure supplement 1A and B*), suggesting that the residual CCDC32 was fully capable of fulfilling this catalytic function.

To further define which stages of CME were dependent on CCDC32, we knocked down CCDC32 in ARPE-HPV cells that stably express eGFP-CLCa and employed quantitative live-cell TIRFM to visualize and analyze CCP dynamics (*Mettlen and Danuser, 2014*). The videos and kymographs of time-lapse imaging showed that CCDC32 depletion resulted in the formation of brighter, static clathrin-coated structures (CCSs) (*Figure 2D and E* and *Figure 2—videos 1 and 2*) that dominate the images. These have been seen under other perturbation conditions (*Aguet et al., 2013*; *Chen et al., 2020*) and reflect the accumulation of CCPs that are either larger, flatter, or both. However, we also noted a subpopulation of dynamic CCPs (arrows, *Figure 2E*) that were visually obscured by bright static CCSs, yet represent the majority of total CCSs detected. Indeed, despite the deceptive nature of the images, which we have previously reported (*Chen et al., 2020*), the percentage of static CCSs (lifetime >150 s), as determined by unbiased quantitative analysis, was only 7.9% (*Figure 2—figure supplement 2*).

To quantify the dynamic behaviors of CCPs, we performed cmeAnalysis (*Aguet et al., 2013*; *Jaqaman et al., 2008*; *Loerke et al., 2011*) and DASC (disassembly asymmetry score classification; *Wang et al., 2020*), which together provide a comprehensive and unbiased characterization of CCP intermediates and CME progression when CCDC32 was depleted. DASC, which measures fluctuations of clathrin-assembly/disassembly to accurately distinguish abortive from productive CCPs (*Wang et al., 2020*), revealed an increased rate of nascent clathrin assembly, reported as the initiation rate of CCSs (*Figure 2F*), but a reduced rate of *bona fide* CCP initiation (*Figure 2G*) and a correspondingly lower percentage of stable, *bona fide* CCPs (CCP%; *Figure 2H*), which are both brighter and longer-lived than abortive CCPs (*Wang et al., 2020*). Together, these data reveal an early defect in the growth and stabilization of nascent clathrin assemblies leading to an increase in the fraction of abortive CCPs. Importantly, the remaining dynamic population of *bona fide* CCPs exhibited shorter lifetimes upon CCDC32 knockdown (*Figure 2I*), indicating their more rapid maturation.

Together, these results demonstrate that CCDC32 is an important endocytic accessory protein involved in CCP stabilization and maturation. Strikingly, despite the profound effects of CCDC32 depletion on CCP dynamics, the efficiency of TfnR uptake was only marginally affected (*Figure 2C*). These paradoxical effects are typical for endocytic accessory proteins and indicative of their functional redundancy and/or the induction of compensatory mechanisms, including in this case, the observed increased rate of CCS assembly (*Figure 2F*) and the more rapid maturation of *bona fide* CCPs (*Figure 2I*; *Aguet et al., 2013*; *Bhave et al., 2020*; *Wang et al., 2020*).

## CCDC32 regulates CCP invagination

During CCP maturation, successful invagination of the clathrin coat and its underlying membrane has been identified as a key step that determines the fate of CCPs (*Aguet et al., 2013*; *Wang et al., 2020*). To determine whether CCDC32 regulates CCP invagination, we first used Epifluorescence (Epi)-TIRF microscopy to measure CCP invagination in live cells (*Figure 3A*). In this approach, time-lapse Epi and TIRF fluorescence signals were near-simultaneously acquired for ARPE-HPV eGFP-CLCa cells (*Figure 3B*) and analyzed using primary (TIRF-channel)/subordinate (Epi-channel) tracking powered

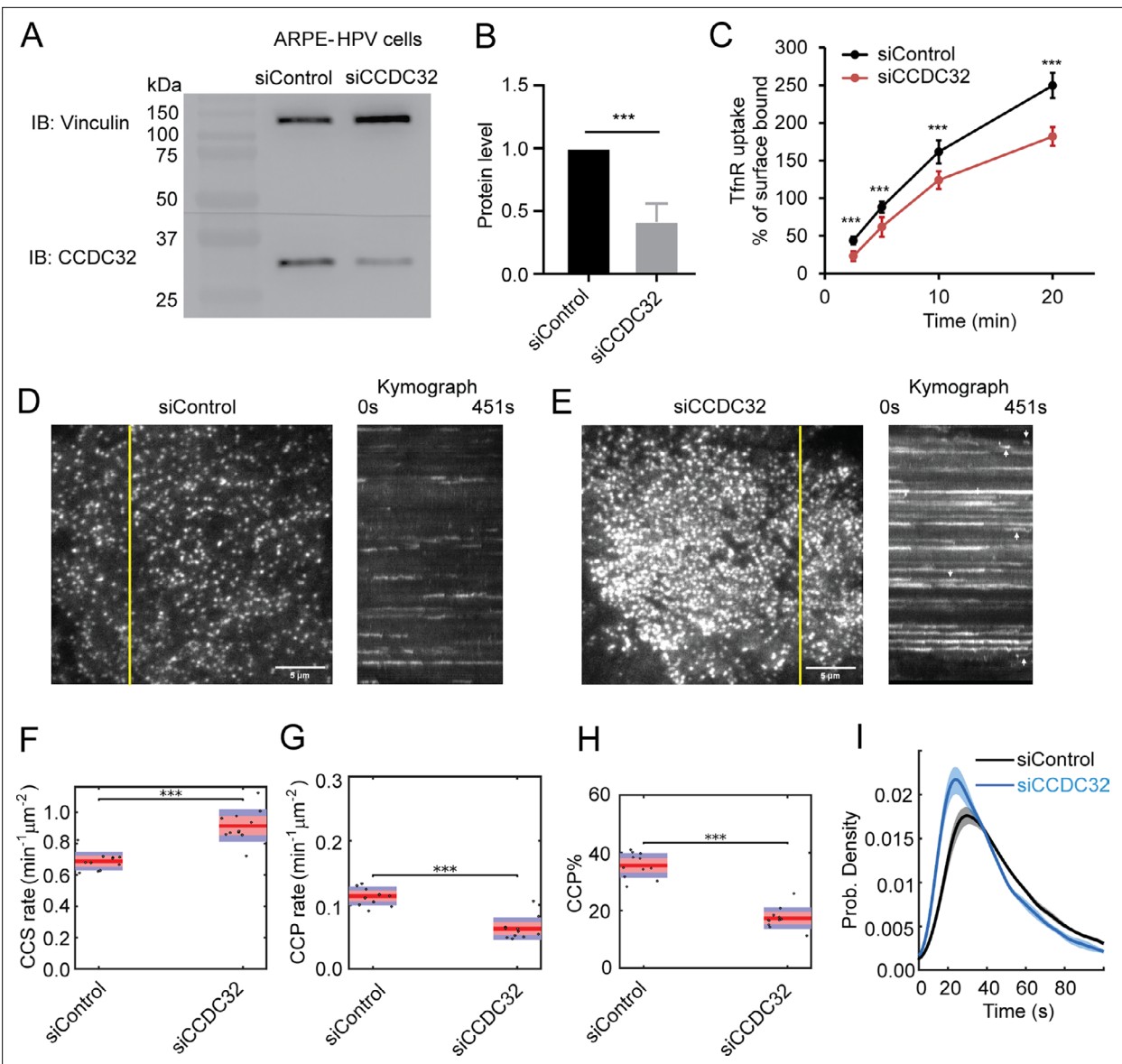

**Figure 2.** CCDC32 depletion inhibits Transferrin Receptor (TfnR) uptake and CCP maturation. (**A**) Immunoblotting (IB) shows efficient CCDC32 knockdown in ARPE-HPV cells by siRNA treatment. (**B**) Quantified knockdown efficiency (~60%) of CCDC32 (n=8). (**C**) Measurements of the uptake efficiency of TfnR (n=8). % of surface bound = Internalized/Surface bound*100%. Error bars in (**B–C**) indicate standard deviations. (**D–E**) Representative single frame images from TIRFM videos (7.5 min/video, 1 frame/s, see *Figure 2—videos 1 and 2*) and corresponding kymographs from region indicated by yellow lines of ARPE-HPV eGFP-CLCa cells treated with (**D**) control siRNA or (**E**) CCDC32 siRNA. Scale bars = 5 μm. (**F–H**) Effect of CCDC32 knockdown on the initiation rates of (**F**) all CCSs and (**G**) *bona fide* CCPs, as well as (**H**) the % of *bona fide* CCPs. Each dot represents a video (n=11). Statistical analysis of the data in (**F–H**) is the Wilcoxon rank sum test, ***, p≤0.001. (**I**) Lifetime distribution of *bona fide* CCPs. Data presented were obtained from a single experiment (n=11 videos for each condition) that is representative of three independent repeats. Number of dynamic tracks analyzed: 125897 for siControl and 105313 for siCCDC32. Shadowed area indicates 95% confidence interval.

The online version of this article includes the following video, source data, and figure supplement(s) for figure 2:

**Source data 1.** Numeric data to generate *Figure 2*.

**Source data 2.** PDF file containing original western blots for *Figure 2A*, indicating the relevant bands and treatments.

**Source data 3.** Original files for western blot analysis displayed in *Figure 2A*.

**Figure supplement 1.** siRNA-mediated knockdown of CCDC32 does not affect AP2 expression level.

**Figure supplement 1—source data 1.** Numeric data to generate *Figure 2—figure supplement 1*.

**Figure supplement 1—source data 2.** PDF file containing original western blots for *Figure 2—figure supplement 1*, indicating the relevant bands and treatments.

*Figure 2 continued on next page*

*Figure 2 continued*

**Figure supplement 1—source data 3.** Original files for western blot analysis displayed in *Figure 2—figure supplement 1*.

**Figure supplement 2.** Fraction of CCPs in lifetime cohorts.

**Figure supplement 2—source data 1.** Numeric data to generate *Figure 2—figure supplement 2*.

**Figure 2—video 1.** Time-lapse TIRFM imaging of ARPE-HPV eGFP-CLCa cells that were treated with control siRNA.
https://elifesciences.org/articles/107039/figures#fig2video1

**Figure 2—video 2.** Time-lapse TIRFM imaging of ARPE-HPV eGFP-CLCa cells that were treated with CCDC32 siRNA.
https://elifesciences.org/articles/107039/figures#fig2video2

by cmeAnalysis (*Wang et al., 2020*). The resulting Epi and TIRF fluorescence intensity traces of CCPs with similar lifetimes were then aligned and averaged to yield intensity cohorts, which were further log-transformed to give average traces of the invagination depth ($\Delta z$) of the CCPs' center-of-mass (*Figure 3A and C*). Here, we chose to present CCPs with lifetimes of 25–35 s because they represent the invagination behavior of the most frequent tracks in control cells (*Figure 2I*). CCDC32 knockdown strongly inhibited CCP invagination (*Figure 3C*), suggesting a key role for CCDC32 in regulating this critical early stage in CME.

To independently validate that CCDC32 regulates CCP invagination, we next examined the clathrin-coated structures (CCSs) at higher resolution using Platinum Replica Electron Microscopy (PREM; *Svitkina, 2017*). Strikingly, CCDC32 knockdown resulted in a substantial increase in the number of flat CCSs (*Figure 4A–C*, pseudo-colored blue in panel B, see also *Figure 4—figure supplement 1A–C*) and a corresponding higher occupancy of CCSs on the PM (*Figure 4E*; *Figure 4—figure supplement 1D*), consistent with TIRFM and Epi-TIRF microscopy observations. While the flat CCSs we detected in CCDC32 knockdown cells were significantly larger than in control cells (*Figure 4D*, mean diameter of 147 nm vs. 127 nm, respectively), they are much smaller than typical long-lived flat clathrin lattices (d≥300 nm; *Grove et al., 2014*). Indeed, the surface area of the flat CCSs that accumulate in CCDC32 KD cells (mean ~1.69 x $10^4$ nm$^2$) remains significantly less than the surface area of an average 100 nm diameter CCV (~3.14 x $10^4$ nm$^2$). Thus, we refer to these structures as 'flat clathrin assemblies' because they are neither curved 'pits' nor large 'lattices'. Rather, the flat clathrin assemblies represent early, likely defective, intermediates in CCP formation. Importantly, while significantly decreased in both number and size, dome-shaped (green) and spherical (orange) CCSs were still detected, likely corresponding to intermediates within the larger subpopulation of dynamic, *bona fide* CCPs able to maintain TfnR uptake (*Figure 4A–D*; *Figure 4—figure supplement 1A–C*). The PREM results provide high-resolution structural data supporting a critical role for CCDC32 in regulating CCP invagination.

## CCDC32 interacts with the AP2 α-appendage domain

Next, we explored the mechanism of CCDC32 recruitment to CCPs. The adaptor AP2, which is essential for CCP and CCV formation, is a heterotetramer consisting of α, β2, μ2, and σ2 subunits (*Figure 5A*). A previous study had reported interactions between overexpressed σ2-mCherry and CCDC32-GFP (*Wainberg et al., 2021*), but had not demonstrated interactions with the native AP2 complex. A more recent study reported that C-terminally tagged CCDC32 interacts with the α:σ2 hemicomplex, but does not interact with the assembled AP2 heterotetramer (*Wan et al., 2024*). To explore this apparent discrepancy, we conducted co-immunoprecipitation (co-IP) experiments from cell lysates of ARPE-HPV cells that stably express a fully functional eGFP-tagged α-subunit of AP2 (*Mino et al., 2020*; AP2-α-eGFP, *Figure 5B*). Mass spectrometry analysis revealed, as expected, the co-IP of all the subunits of the AP2 complex, as well as well-known AP2 binding proteins (i.e. EPS15, NECAP2, AAK1, and AAGAB; *Mettlen et al., 2018*). Notably, CCDC32 was also efficiently co-immunoprecipitated with intact AP2 (*Figure 5C*; *Table 1*). However, this experiment does not definitively rule out the possibility that CCDC32 only interacts with a small population of immature α:σ2 hemicomplexes that might be present in the lysate. Therefore, we performed co-IP experiments in cell lysates from ARPE-HPV cells stably expressing eGFP-CCDC32(FL) (*Figure 5D*). Western blotting showed that the α subunit (*Figure 5E and F*), together with each of the AP2 complex subunits (*Figure 5—figure supplement 1*), was efficiently co-precipitated with eGFP-CCDC32(FL), but not by GFP (*Figure 5D–F*). Thus, we conclude that CCDC32 interacts with the mature AP2 complex. The discrepancy with previous findings may be due to the location of the eGFP tag (see Discussion).

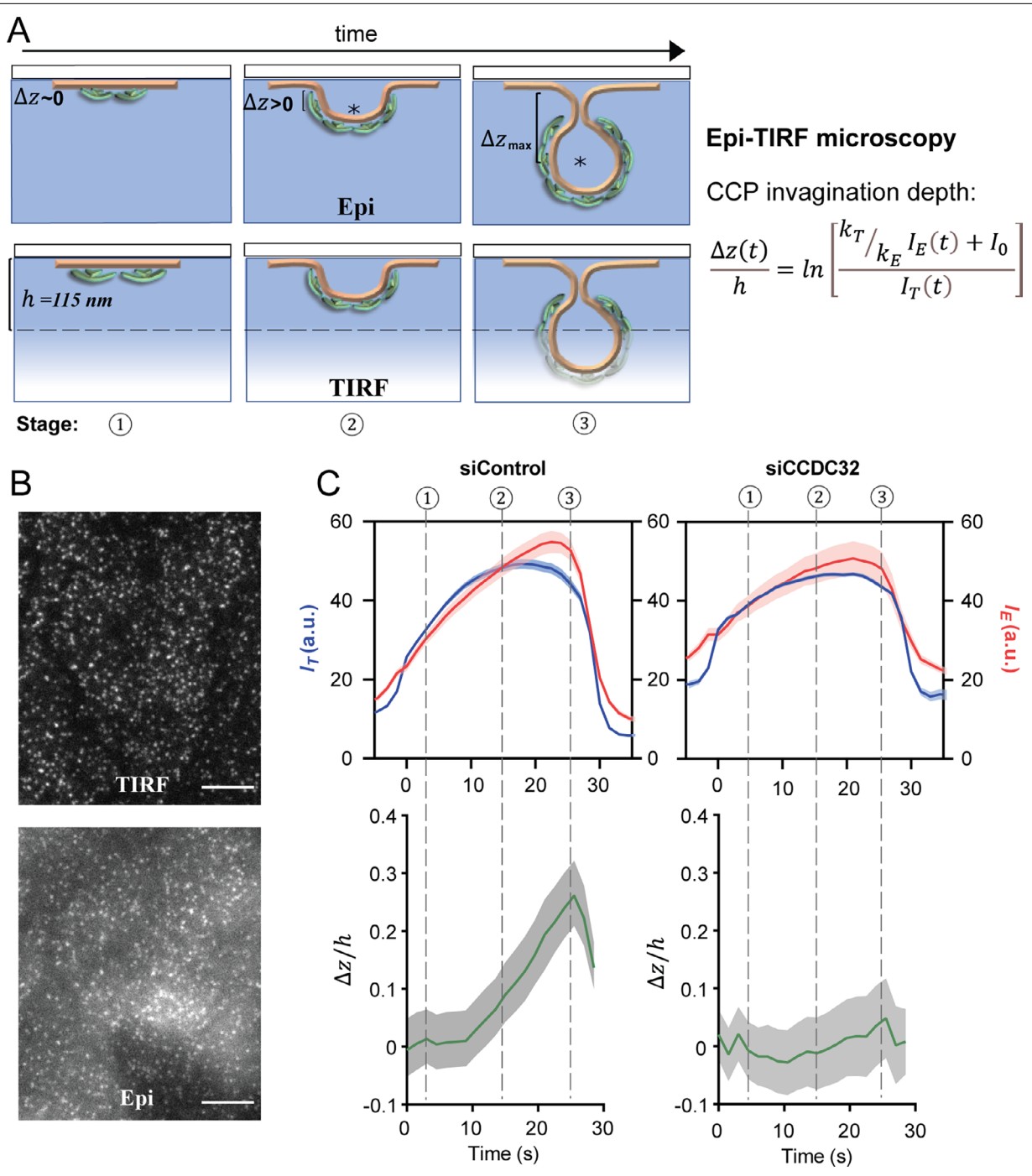

**Figure 3.** CCDC32 depletion inhibits CCP invagination. (**A**) Scheme of Epi-TIRF microscopy for measuring the invagination of CCPs using primary/subordinate tracking. $\Delta z\,(t)$ denotes the invagination depth of CCPs over time. $I_E$ and $I_T$ denotes the cohort-averaged fluorescence intensity from Epi and TIRF channels, respectively. $k_E$ and $k_T$ are the initial growth rate for the Epi and TIRF channel signals, respectively. $I_0$ is an additive correction factor. '*' indicates the mass center of clathrin coat. $h$=115 nm is the evanescent depth of TIRF field, see more detail in *Saffarian and Kirchhausen, 2008*; *Wang et al., 2020*. (**B**) Representative Epi and TIRF microscopy images. Scale bars = 10 µm. (**C**) Epi-TIRF microscopy analysis shows that knockdown of CCDC32 strongly inhibited CCP invagination. Top: Cohort-averaged CCP fluorescence intensity traces from Epi and TIRF channels; bottom: calculated $\Delta z\,(t)\,/h$ curves. Data presented were obtained from n=12 videos for each condition. Number of CCP tracks analyzed to obtain the $\Delta z/h$ curves: 5998 for siControl and 4418 for siCCDC32. Shadowed area indicates 95% confidential interval.

The online version of this article includes the following source data for figure 3:

**Source data 1.** Numerical data to generate *Figure 3*.

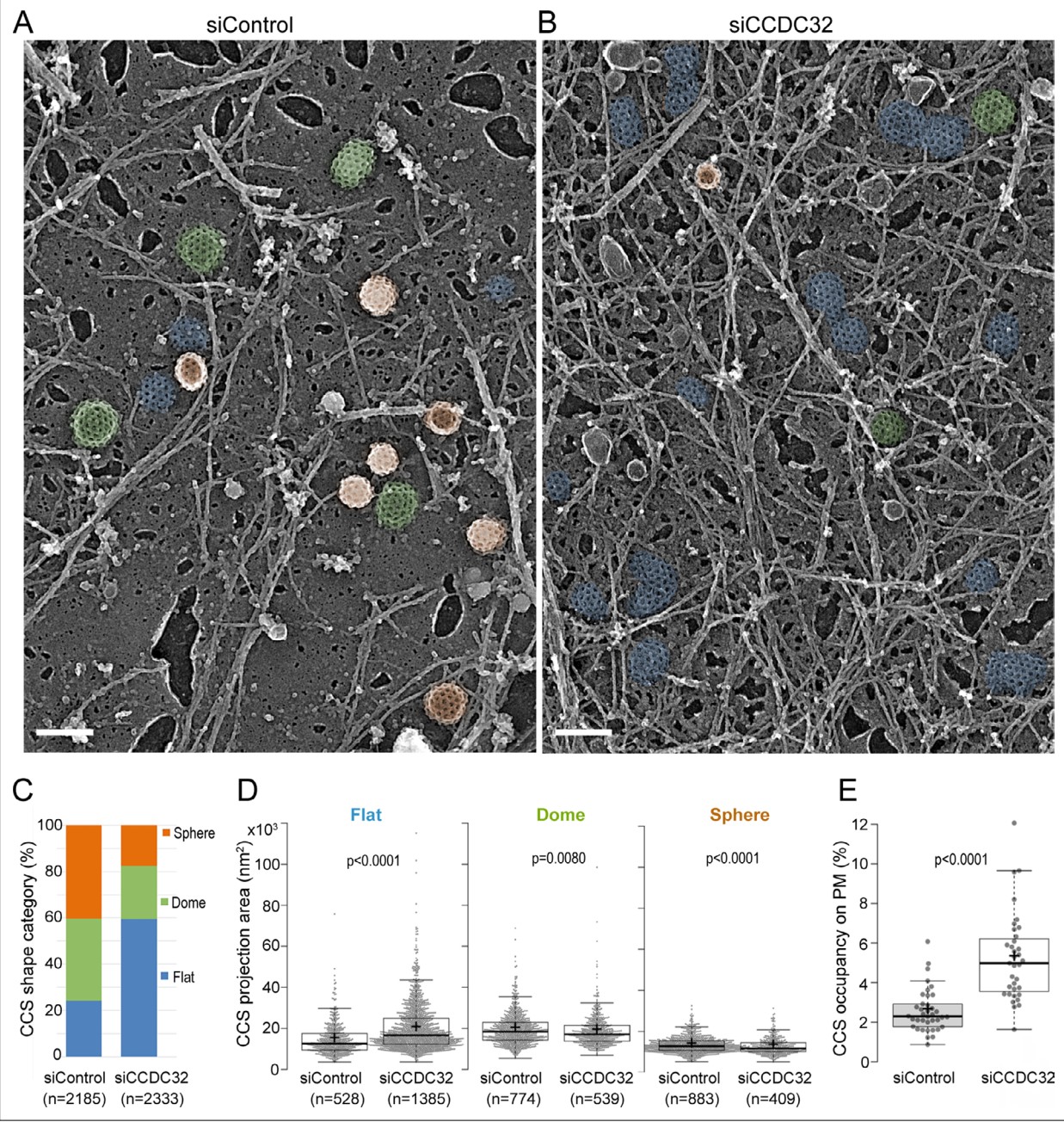

**Figure 4.** Flat clathrin lattices accumulate in cells depleted of CCDC32. (**A–B**) Representative PREM images of ARPE19 cells treated with (**A**) control siRNA or (**B**) CCDC32 siRNA showing flat (blue), dome-shaped (green), and spherical (orange) CCSs. Scale bars = 200 nm. (**C–E**) Quantification of the clathrin-coated structures (CCSs) by (**C**) shape category, (**D**) the CCS projection area, and (**E**) the CCS occupancy on the plasma membrane (PM). Each dot in (**E**) represents an individual fragment of the PM; the number of cell membrane fragments analyzed is 38 for siControl cells and 35 for siCCDC32 cells from two independent experiments, n=number of CCS. Statistical tests were performed using Mann-Whitney test (**D**) or unpaired t-test (**E**). For the Box and whisker plots in (**D**) and (**E**), the box extends from the 25th to 75th percentiles, the line in the middle of the box is plotted at the median, and the '+' indicates the mean.

The online version of this article includes the following source data and figure supplement(s) for figure 4:

**Source data 1.** Numerical data to generate *Figure 4*.

**Figure supplement 1.** CCDC32 knockdown shifts the shapes of clathrin-coated structures.

**Figure supplement 1—source data 1.** Numerical data to generate *Figure 4—figure supplement 1*.

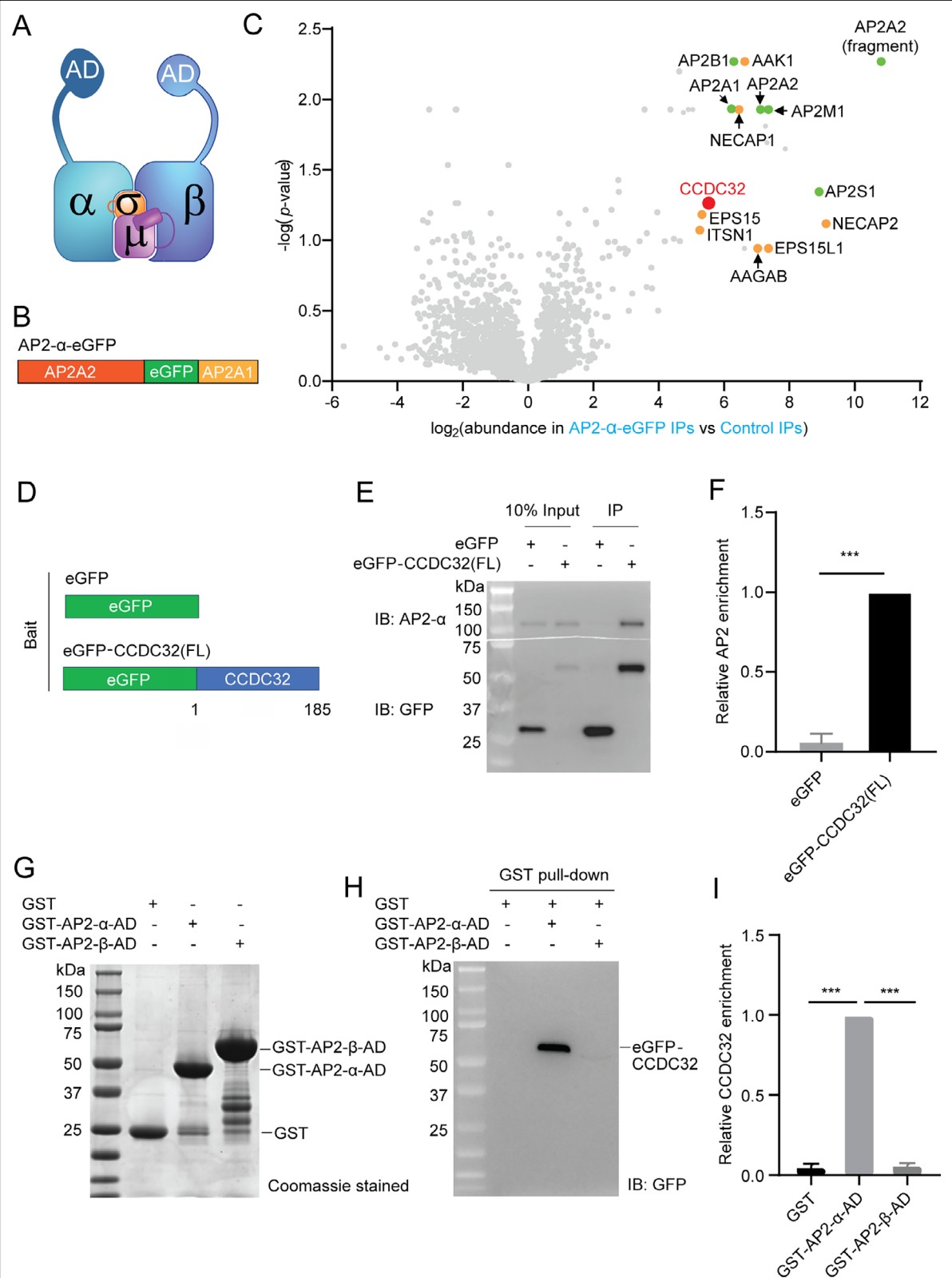

**Figure 5.** CCDC32 interacts with AP2 through the α appendage domain. (**A**) AP2 structure. AD: appendage domain. (**B**) Domain structure of a fully-functional α-subunit bearing an internal eGFP tag inserted in the unstructured hinge region. (**C**) Volcano plot after mass spectrometry analysis of the immunoprecipitation (IP) of eGFP from naive ARPE-HPV cells (control IPs) or ARPE-HPV cells that stably express AP2-α-eGFP (AP2-α-eGFP IPs) using anti-GFP beads. Green: AP2 subunits; orange: known AP2 interactors. Source data for the volcano plot is available as *Figure 5—source data 1*. (**D–F**)

*Figure 5 continued on next page*

*Figure 5 continued*

IP of eGFP from ARPE-HPV cells that stably express eGFP or eGFP-CCDC32(FL) using anti-GFP magnetic beads. (**D**) The domain structures of eGFP and eGFP-CCDC32(FL). (**E**) Representative immunoblotting result of n=3 IP samples. (**F**) Relative AP2 enrichments quantified from immunoblotting results. (**G–I**) GST pull-down assays. (**G**) Coomassie blue stained SDS-page gel of purified GST, GST-AP2-α-AD, and GST-AP2-β-AD. (**H**) Representative western blot result of n=3 GST pull-down assay from ARPE-HPV eGFP-CCDC32(FL) cell lysate using purified GST, GST-AP2-α-AD, or GST-AP2-β-AD. The amount of bait GST-proteins is shown in (**G**), and the pulled down eGFP-CCDC32 was detected by immunoblotting (IB) of GFP. (**I**) Quantification of immunoblots of the relative enrichment of CCDC32. Error bars in (**F**) and (**I**) are SD (n=3). Two-tailed student's t-test: ***, p≤0.001.

The online version of this article includes the following source data and figure supplement(s) for figure 5:

**Source data 1.** Original MassSpec results displayed in 5 C.

**Source data 2.** Numerical data to generate *Figure 5F and I*.

**Source data 3.** PDF file containing original western blots for *Figure 5E, G and H*, indicating the relevant bands and treatments.

**Source data 4.** Original files for western blot analysis displayed in *Figure 5E, G and H*.

**Figure supplement 1.** All the four subunits of intact AP2 efficiently co-IP with eGFP-CCDC32(FL).

**Figure supplement 1—source data 1.** PDF file containing original western blots for *Figure 5—figure supplement 1*, indicating the relevant bands and treatments.

**Figure supplement 1—source data 2.** Original files for western blot analysis displayed in *Figure 5—figure supplement 1*.

Most endocytic accessory proteins interact with AP2 via the appendage domains of the α and β2 subunits (*Praefcke et al., 2004*; *Schmid et al., 2006*; *Figure 5A*, α-AD and β-AD). Therefore, we expressed and purified GST-tagged AP2 α-AD and β-AD (*Figure 5G*) and used GST pull-down assays to test which might interact with CCDC32. GST-AP2-α-AD, but not GST-AP2-β-AD, was capable of pulling down CCDC32 from ARPE-HPV eGFP-CCDC32(FL) cell lysate (*Figure 5H and I*). Thus, CCDC32 can interact with AP2 via the α-AD.

**Table 1.** ID and abundances of key proteins detected in IP samples by Mass Spectroscopy analysis.
#1-#3: 3 independent repeats of control IPs; #4-#6: 3 independent repeats of AP2-α-eGFP IPs.

| | Abundance | | | | | |
| --- | --- | --- | --- | --- | --- | --- |
| | **Control IPs** | | | **AP2-α-eGFP IPs** | | |
| Gene | #1 | #2 | #3 | #4 | #5 | #6 |
| AP2A1 | 2.E+06 | 1.E+06 | 6.E+05 | 1.E+08 | 1.E+08 | 2.E+08 |
| AP2A2 | 4.E+06 | 1.E+06 | 7.E+05 | 3.E+08 | 3.E+08 | 3.E+08 |
| AP2B1 | 7.E+06 | 6.E+06 | 3.E+06 | 7.E+08 | 6.E+08 | 6.E+08 |
| AP2M1 | 1.E+06 | 8.E+05 | 2.E+05 | 2.E+08 | 1.E+08 | 2.E+08 |
| AP2S1 | 1.E+05 | 5.E+04 | 9.E+03 | 2.E+07 | 2.E+07 | 5.E+07 |
| CCDC32 | 0.E+00 | 0.E+00 | 0.E+00 | 3.E+06 | 2.E+06 | 2.E+06 |
| AAK1 | 7.E+04 | 1.E+05 | 3.E+04 | 2.E+07 | 1.E+07 | 2.E+07 |
| EPS15 | 2.E+06 | 3.E+05 | 2.E+05 | 3.E+07 | 3.E+07 | 3.E+07 |
| EPS15L | 3.E+05 | 1.E+04 | 2.E+04 | 1.E+07 | 1.E+07 | 1.E+07 |
| FCHO2 | 4.E+04 | 3.E+04 | 2.E+04 | 1.E+06 | 1.E+06 | 1.E+06 |
| ITSN1 | 1.E+04 | 0.E+00 | 0.E+00 | 1.E+06 | 9.E+05 | 7.E+05 |
| ITSN2 | 0.E+00 | 0.E+00 | 0.E+00 | 0.E+00 | 0.E+00 | 8.E+04 |
| NECAP1 | 9.E+04 | 8.E+04 | 3.E+04 | 9.E+06 | 7.E+06 | 1.E+07 |
| NECAP2 | 1.E+05 | 7.E+03 | 1.E+04 | 2.E+07 | 2.E+07 | 2.E+07 |
| AAGAB | 9.E+00 | 0.E+00 | 0.E+00 | 3.E+06 | 4.E+06 | 7.E+06 |

## Identification of a CCDC32 binding site for AP2 interactions

We next identified a region on CCDC32 responsible for AP2 binding. Structural predictions of CCDC32 made by AlphaFold 3.0 show it to be a mainly unstructured protein with several isolated α-helices (*Figure 6—figure supplement 1A* and B), which likely led to its misnomer as a coiled-coil domain containing protein. Indeed, even when modeled as a dimer or trimer, we found no evidence of coiled-coil interactions between the α-helices (data not shown). Moreover, endogenous CCDC32 did not co-immuno-precipitate with eGFP-CCDC32 from ARPE-HPV eGFP-CCDC32(FL) cell lysate (*Figure 6—figure supplement 1C*), indicating that the protein is a monomer in vivo. Nonetheless, we focused first on a strongly predicted α-helix encoded by aa78-98 and located in the middle of CCDC32 and engineered a CCDC32 construct lacking this α-helix (Δ78–98, *Figure 6A*; *Figure 6—figure supplement 1A* and B). The corresponding siRNA-resistant CCDC32(Δ78–98) mutant was stably expressed in ARPE-HPV cells at comparable expression levels to eGFP-CCDC32(FL) (*Figure 1—figure supplement 1A*). As previously shown (*Figure 5—figure supplement 1*), all four subunits of AP2 efficiently co-IP with full-length CCDC32 from cell lysates (*Figure 6B and C*). Interestingly, the co-IPs of both α and σ2 subunits with CCDC32(Δ78–98) were greatly reduced (*Figure 6B and C*); whereas, unexpectedly, the ability of CCDC32(Δ78–98) to interact with β2:μ2 was unaffected. These data are partially consistent with the results of *Wan et al., 2024*, who reported interactions with the α:σ2 hemicomplex and with μ2, but not with β2. Given the efficiency and selectivity of our co-IP, our data also suggests that the mature AP2 complex exists in a dynamic equilibrium between α:σ2 and β2:μ2 hemicomplexes. Indeed, it has previously been reported that β2:μ2 hemicomplexes can partially support synaptic vesicle recycling in *C. elegans* bearing null mutations in the α-subunit of AP2 (*Gu et al., 2013*). These findings demonstrate that aa78-98 are essential for CCDC32 interactions with both mature AP2 complexes and the α:σ2 hemicomplex.

Disrupting AP2 interactions severely impaired the recruitment of CCDC32(Δ78–98) to CCPs (*Figure 6D and E*), indicating that CCDC32 is likely recruited to CCPs through its interactions with AP2. Notably, unlike CCDC32(FL), CCDC32(Δ78–98) was unable to rescue either the uptake of the CME cargo, TfnR (*Figure 6F*) or the stabilization of CCPs (*Figure 6G*). Together, these results establish that CCDC32, a previously uncharacterized endocytic accessory protein, plays a critical role in regulating CCP stabilization and invagination.

## Disease-causing nonsense mutation in CCDC32 loses AP2 interaction capacity and inhibits CME

Loss-of-function nonsense mutations in CCDC32 have been reported to result in CFNDS (*Abdalla et al., 2022*; *Harel et al., 2020*); however, the disease-causing mechanism remains unknown. The identified frameshift mutations result in premature termination and truncation of the protein at residues 10, 55, or 81 (*Harel et al., 2020*). Based on our findings, we hypothesize that these C-terminally truncated mutants, all of which lack the α-helix encoded by residues 78–98, will be defective in AP2 binding and unable to function in CME. To test this hypothesis, we generated an siRNA-resistant eGFP-CCDC32(1-54) construct (*Figure 7A*) based on the clinical report (*Harel et al., 2020*), and stably expressed this truncated CCDC32 mutant in ARPE-HPV cells (*Figure 1—figure supplement 1A*).

As predicted, AP2 did not co-IP with CCDC32(1-54) from the cell lysate (*Figure 7B and C*), confirming that the disease-causing mutation in CCDC32 abolishes its interactions with AP2. Correspondingly, we could not detect CCDC32(1-54) recruitment to CCPs in dual channel TIRFM imaging (*Figure 7D and E*), consistent with CCP recruitment of CCDC32 being dependent on CCDC32-AP2 interactions. Interestingly, we also noticed that the level of diffuse plasma membrane binding detected by TIRFM decreased to a greater extent than that seen for CCDC32(Δ78–98), now corresponding to background, eGFP only levels (see *Figure 7—figure supplement 1* for direct comparison). These data suggest that the C-terminus of CCDC32 is required for binding to the inner surface of the plasma membrane. Finally, we observed that expression of eGFP-CCDC32(1-54), after siRNA-mediated knockdown of endogenous CCDC32, was unable to rescue TfnR uptake efficiency (*Figure 7F*) or CCP stabilization (*Figure 7G*). These findings show that this loss-of-function nonsense mutation in CCDC32 abolishes its interactions with AP2 and inhibits CME, likely contributing to the development of CFNDS.

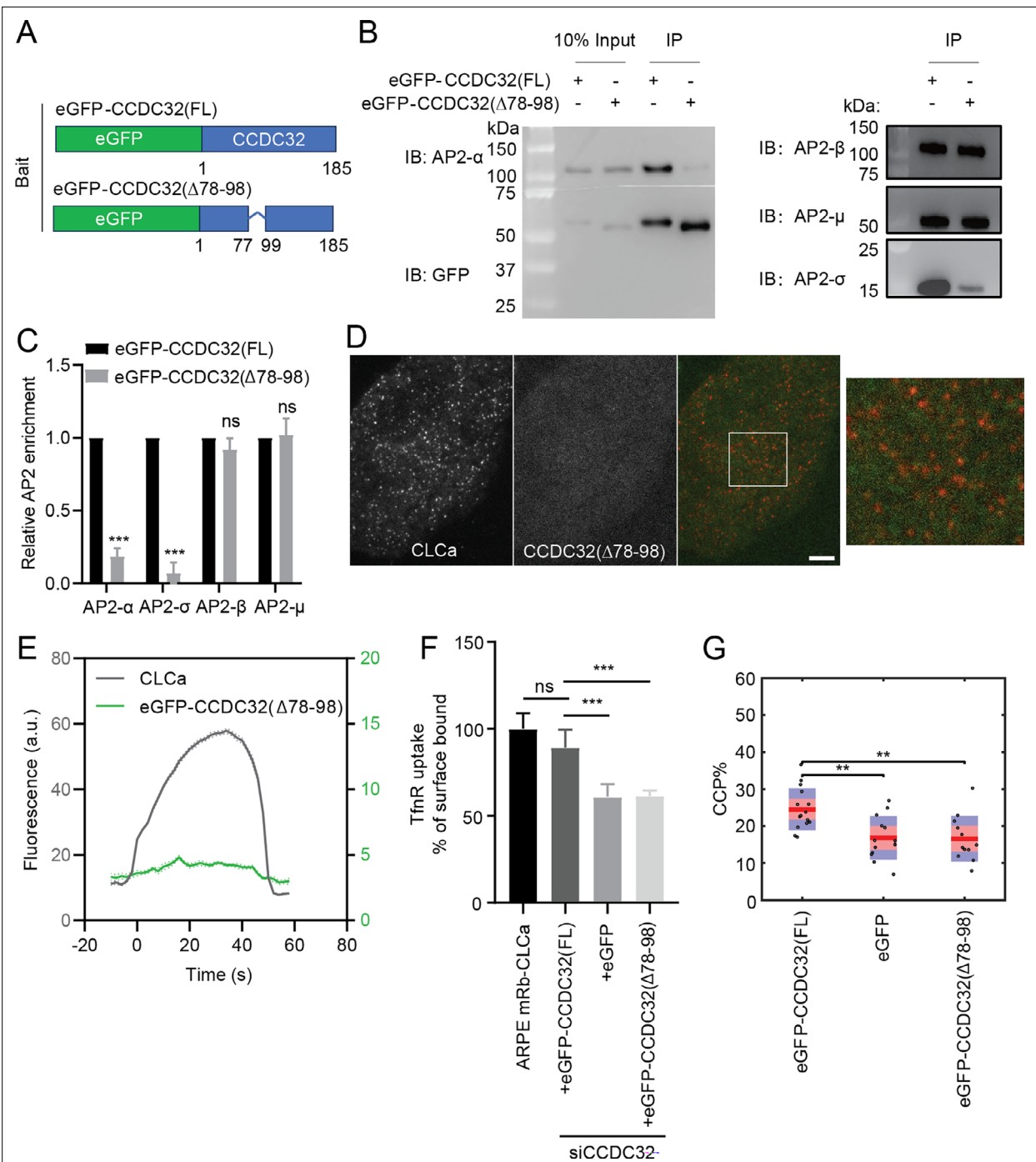

**Figure 6.** A central α-helix in CCDC32 mediates CCDC32-AP2 interactions. (**A–C**) IP of eGFP from ARPE-HPV cells that stably express eGFP-CCDC32(FL) or eGFP-CCDC32(Δ78–98) using anti-GFP beads. (**A**) The domain structures of eGFP-CCDC32(FL) and eGFP-CCDC32(Δ78–98). (**B**) Representative immunoblotting result of n=3 IP samples. (**C**) Relative AP2 enrichments that were quantified from (**B**). Error bars indicate standard deviations. (**D**) Representative TIRFM images of ARPE-HPV cells that stably express mRuby-CLCa and eGFP-CCDC32(Δ78–98). White ROI is magnified on the right. Scale bar = 5 μm. (**E**) Cohort-averaged fluorescence intensity traces of CCPs and CCP-enriched eGFP-CCDC32(Δ78–98). Number of tracks analyzed: 37892. (**F–G**) (**F**) TfnR uptake efficiency and (**G**) CCP% of ARPE mRb-CLCa cells that stably express eGFP-CCDC32(WT), eGFP, or eGFP-CCDC32(Δ78–98), and with siRNA-mediated knockdown of endogenous CCDC32. Each dot in (**G**) represents a video. Statistical analysis of the data in (**F**) is two-tailed student's t-test: ns, not significant; ***, p≤0.001. Statistical analysis of the data in (**G**) is the Wilcoxon rank sum test, **, p≤0.01.

The online version of this article includes the following source data and figure supplement(s) for figure 6:

**Source data 1.** Numerical data to generate *Figure 6*.

*Figure 6 continued on next page*

*Figure 6 continued*

**Source data 2.** PDF file containing original western blots for *Figure 6B*, indicating the relevant bands and treatments.

**Source data 3.** Original files for western blot analysis displayed in *Figure 6B*.

**Figure supplement 1.** The sequence and structure of CCDC32.

**Figure supplement 1—source data 1.** PDF file containing original western blots for *Figure 6—figure supplement 1C*, indicating the relevant bands and treatments.

**Figure supplement 1—source data 2.** Original files for western blot analysis displayed in *Figure 6—figure supplement 1C*.

## Discussion

CCDC32 was identified through genetic studies, originally in a yeast 2-hybrid screen for proteins interacting with annexin A2 (*Li et al., 2011*) and subsequently via whole exome sequencing to identify mutations associated with cranio-facio-neurodevelopmental syndrome (CFNDS) (*Abdalla et al., 2022*; *Harel et al., 2020*), neither of which provided insight into its cellular function. Subsequent bioinformatic analysis of co-essential modules linked CCDC32 with the AP2 adaptor complex, provided evidence for its interaction with AP2, reported colocalization of CCDC32 with AP2 at CCPs, and demonstrated a role in CME (*Wainberg et al., 2021*). A more recent report has suggested that CCDC32 functions as a chaperone, essential for the assembly of mature AP2 heterotetrameric complexes (*Wan et al., 2024*), but that CCDC32 neither binds to the mature AP2 complex nor colocalizes with CCPs.

Here, we show that CCDC32 binds to intact AP2 complexes and that this interaction is required for its recruitment to CCPs. Our data demonstrate that CCDC32 plays a critical role at early stages of CME, dependent on its recruitment to CCPs. Depletion of CCDC32 results in a pronounced defect in CCP invagination, a decrease in the rate of formation and percentage of stabilized nascent *bona fide* CCPs and an accumulation of flat clathrin assemblies, unstable intermediates in CCP formation (*Figure 8*). Despite these profound alterations in CCP dynamics, CME itself, as measured by TfnR internalization efficiency, is only partially inhibited. We speculate that this mild endocytic defect reflects the plasticity and resilience of CME. Indeed, we detect two potential compensatory mechanisms that occur upon depletion of CCDC32, namely an increase in the rate of CCS assembly and a decrease in the lifetimes of dynamic *bona fide* CCPs (i.e. an increased rate of CCP maturation; *Figure 8*).

While individual depletion of other endocytic accessory proteins has been reported to result in one or more of these phenotypes, to our knowledge, this combination of phenotypes has not been detected. Instead, either CCP assembly is reduced (*Henne et al., 2010*; *Umasankar et al., 2014*) or CCPs accumulate at later stages of invagination (*McMahon and Boucrot, 2011*; *Messa et al., 2014*). This combination of phenotypes has, however, been observed previously in cells expressing a truncated mutant of AP2 lacking the α-AD (*Aguet et al., 2013*). These cells also exhibited: (i) near normal Tfn uptake, (ii) an accumulation of flat clathrin assemblies, which are rapidly turned over, (iii) an increase in the rate of nascent CCS initiation, (iv) impaired stabilization of dynamic CCPs and, (v) a decrease in productive CCP lifetimes. We speculate that loss of CCDC32 recruitment could have accounted for these unique phenotypes.

Although the mechanism of CCP invagination has been an important topic for decades, which endocytic accessory proteins (if any) regulate CCP invagination and (if so) how, has remained unclear. The depletion of CCDC32 in cells strongly inhibited the formation of dome-shaped and spherical CCSs and instead resulted in the accumulation of flat clathrin assemblies. As the area of these clathrin lattices was insufficient to form a complete CCV, they likely represent defective CCP assembly intermediates that are rapidly turned over as abortive CCPs. This experimentally observed requirement of CCDC32 in CCP invagination is consistent with previous findings that clathrin assembly alone was not sufficient to induce CCP invagination in cells (*Aguet et al., 2013*), potentially due to the reported flexibility of clathrin coats (*Ferguson et al., 2008*; *Tagiltsev et al., 2021*). The mechanism by which CCDC32 promotes CCP invagination remains to be determined, but other domains of the protein may bind to and stabilize the clathrin coat, or the predicted largely disordered structure of CCDC32 (*Figure 6—figure supplement 1*) may generate membrane curvature by molecular crowding (*Busch et al., 2015*).

We localized the AP2 binding site on CCDC32, which is essential for CCDC32-AP2 interactions in vivo, to a short, predicted alpha-helical region (aa78-98). Interestingly, this sequence, LASLEKKLRRIK GLNQEVTSKD, does not encode any of the known AP2 α-AD binding motifs (e.g. DxF/W, FxDxF,

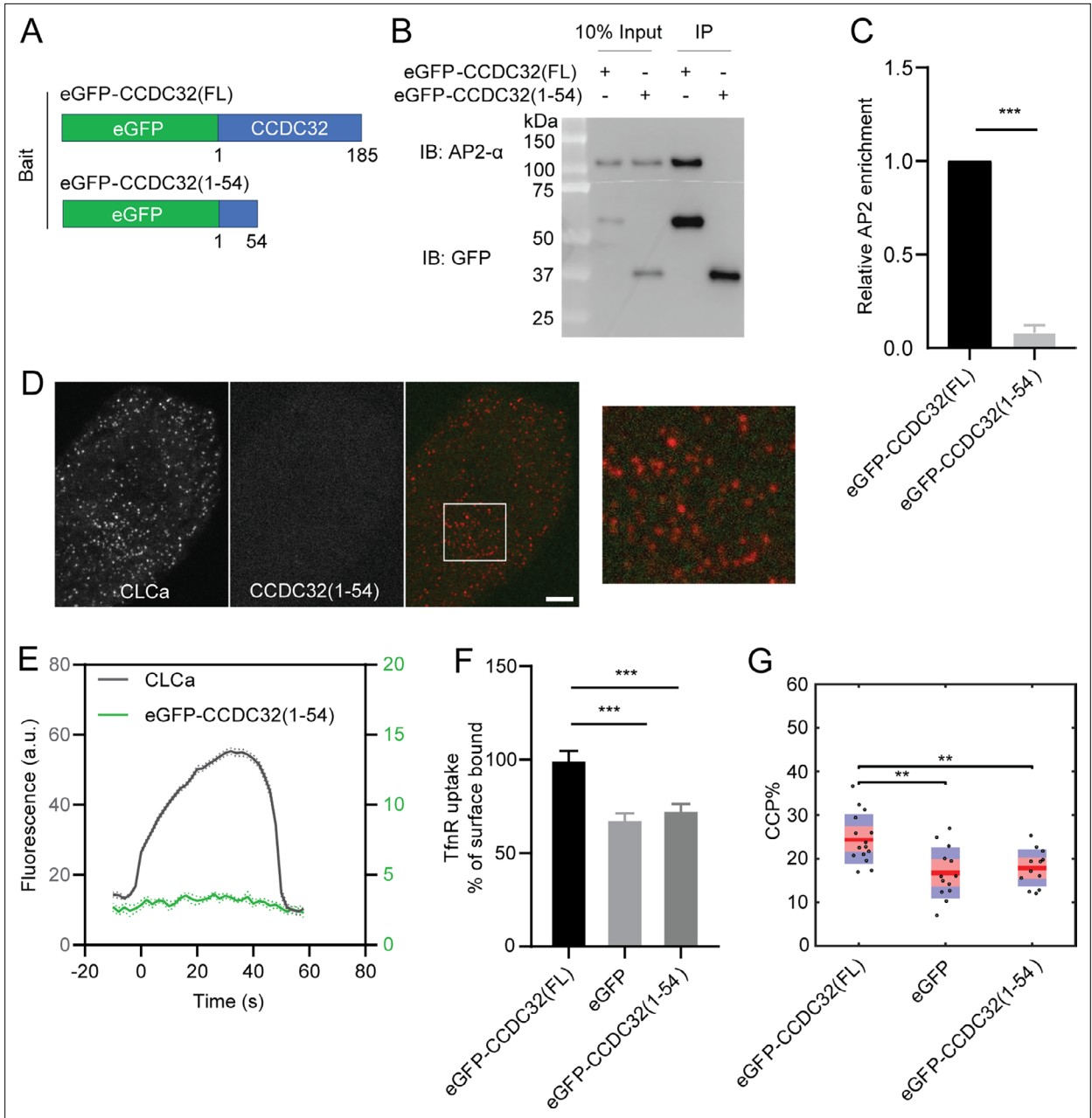

**Figure 7.** Disease-causing nonsense mutation in CCDC32 loses AP2 interaction capacity and inhibits CME. (**A–C**) IP of eGFP from ARPE-HPV cells that stably express eGFP-CCDC32(FL) or eGFP-CCDC32(1-54) using anti-GFP magnetic beads. (**A**) The domain structures of eGFP-CCDC32(FL) and eGFP-CCDC32(1-54). (**B**) Representative immunoblotting result of n=3 IP samples. (**C**) Relative AP2 enrichments quantified from immunoblotting results. (**D**) Representative TIRFM images of ARPE-HPV cells that stably express mRuby-CLCa and eGFP-CCDC32(1-54). White ROI is magnified on the right. Scale bar = 5 μm. (**E**) Cohort-averaged fluorescence intensity traces of CCPs and CCP-enriched eGFP-CCDC32(1-54). Number of tracks analyzed: 28658. (**F–G**) (**F**) TfnR uptake efficiency and (**G**) CCP% of ARPE-HPV cells that stably express eGFP-CCDC32(FL), eGFP, or eGFP-CCDC32(1-54), and with siRNA-mediated knockdown of endogenous CCDC32. Each dot in (**G**) represents a video. Statistical analysis of the data in (**F**) is two-tailed student's t-test: ***, p≤0.001. Statistical analysis of the data in (**G**) is the Wilcoxon rank sum test, **, p≤0.01.

The online version of this article includes the following source data and figure supplement(s) for figure 7:

**Source data 1.** Numerical data to generate *Figure 7*.

**Source data 2.** PDF file containing original western blots for *Figure 7B*, indicating the relevant bands and treatments.

**Source data 3.** Original files for western blot analysis displayed in *Figure 7B*.

**Figure supplement 1.** A collection of representative TIRFM images from the main text showing cells that stably express the same amount of eGFP-CCDC32(FL), eGFP, eGFP-CCDC32(Δ78–98), and eGFP-CCDC32(1-54).

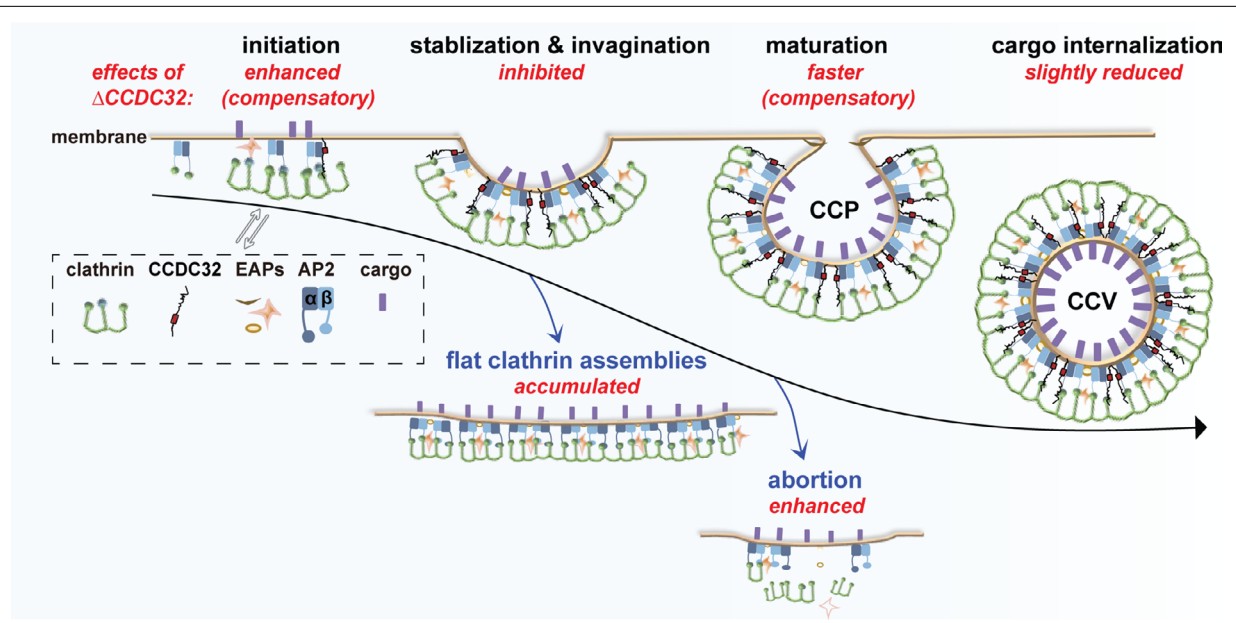

**Figure 8.** Cartoon illustration of CCDC32-AP2 interactions that regulate CME. Interactions between residues 78–98, corresponding to a central α-helix of CCDC32, and AP2 are essential for recruitment of CCDC32 to CCPs. Depleting CCDC32 (ΔCCDC32) inhibits CCP stabilization and invagination, resulting in enhanced CCS abortion and accumulated flat clathrin assembly intermediates. As a compensation effect, CCS initiation is enhanced and CCP maturation is faster, resulting in only slightly reduced cargo internalization. EAPs: endocytic accessory proteins.

WxxF/W, FxxFxxL) commonly shared amongst other AP2 interacting partners (*Olesen et al., 2008*). However, we identified two canonical α-AD binding motifs in CCDC32 ([17]DLW[19] and [39]FSDSF[43]), both of which can be docked on the AP2 appendage domain with high confidence using AlphaFold 3.0 (*Figure 6—figure supplement 1D* and E). While these binding motifs may have been sufficient for in vitro interactions with α-AD, they were not sufficient for AP2 binding in vivo. Unexpectedly, AlphaFold 3.0 modeling predicts, with high confidence, that the α-helical region we identified as essential for AP2 binding in vivo docks to an alpha helix formed by aa418-436 of the α-subunit (*Figure 6—figure supplement 1D* and F), which is not encoded in our α-AD construct. This site is located in the α-trunk and buried with the AP2 core domain (*Collins et al., 2002*). However, it becomes accessible when the C-terminal domain of μ2 adopts the open conformation triggered by cargo binding (*Jackson et al., 2010*). Thus, we speculate that CCDC32 may preferentially interact with membrane-bound AP2. Interestingly, while co-IP with CCDC32(Δ78–99) no longer pulled down the intact AP2 complex or the α:σ2 hemicomplex, it remained capable of pulling down the β2:μ2 hemicomplex. This result has a number of interesting implications: first, that the extended full-length CCDC32 has multiple points of interaction with AP2 and its subunits, that will require further studies to structurally and functionally define; and second, that the AP2 complex is less stable than previously assumed (i.e. the two hemicomplexes must be in flux). Indeed, it has been shown that β2:μ2 hemicomplexes can partially support synaptic vesicle recycling in *C. elegans* bearing null mutations of the α-subunit (*Gu et al., 2013*). Further structural studies will be needed to identify the full extent of CCDC32 interactions with the AP2 complex.

While this paper was under review, it was reported that CCDC32 functions together with AAGAB, as an essential chaperone for AP2 assembly (*Wan et al., 2024*). While there are discrepancies, our results are not incompatible with their findings. Thus, it is possible, even likely, that the ~40% residual CCDC32 present after siRNA knockdown may be sufficient to fulfill its catalytic chaperone activity in facilitating AP2 complex assembly, but not its presumed structural role in regulating early stages of CME. The inability of *Wan et al., 2024* to detect CCDC32 binding to mature AP2 complexes or β2:μ2 hemicomplexes or its recruitment to CCPs may reflect a perturbed function of the C-terminally tagged construct. Indeed, their images show the formation of large (>100 nm), cytoplasmic puncta of CCDC32-GFP, which might be reflective of protein aggregation. Moreover, our data show a role for the C-terminus in diffuse plasma membrane binding. Finally, we find that the Δ78–98 mutant retains its ability to bind the β2:μ2 hemicomplex, but loses its ability to bind α:σ2

hemicomplex. Thus, the most parsimonious conclusion is that CCDC32 is multifunctional, acting catalytically to facilitate AP2 assembly and, after recruitment to CCPs, to regulate early stages of CME. Further studies are needed to fully elucidate the mechanisms governing each of these functions.

Our results provide an explanation for how the clinically observed loss-of-function mutations in CCDC32 result in the development of CFNDS. As we have shown, the disease-associated CCDC32 loss-of-function mutants are not capable of interacting with intact AP2, thus losing their AP2 regulation capacity. Our results suggest that the inability to bind mature AP2 and hence to be recruited to nascent CCSs inhibits critical early stages of CME and contributes to the development of CFNDS. While we cannot rule out nonsense-mediated decay and resulting loss of expression of the truncated human mutations (*Lykke-Andersen and Jensen, 2015*), it is interesting that the clinical features of CCDC32 loss-of-function are similar to those resulting from AP2 loss-of-function mutations (*Arrigo and Lin, 2021*; *Gorvin et al., 2017*; *Helbig et al., 2019*; *Jung et al., 2015*; *Li et al., 2018*; *Li et al., 2010*), supporting that CCDC32 functions through AP2. However, we suggest that the disease-associated CFNDS mutants are hypomorphic, as the complete loss of AP2 complexes, which would result from a complete loss of the CCDC32 protein (*Wan et al., 2024*), has been shown to be embryonic lethal in *Drosophila* (*González-Gaitán and Jäckle, 1997*) and zebrafish (*Umasankar et al., 2012*).

In summary, our study identified CCDC32 as an important endocytic accessory protein that regulates CCP stabilization and invagination via its interactions with AP2. Future work needs to address the multifunctional interactions of CCDC32 with AP2 and to define exactly how CCDC32 enhances curvature generation and coat stabilization.

# Materials and methods

## Plasmids

The eGFP-CCDC32(FL, human) cDNA in a pEGFP-C1 vector was purchased from Addgene (#110505) and then mutated to be siRNA resistant, which retained amino acid sequence (#98–102) while modifying the nucleotide sequence. Next, aa78-98 was deleted from siRNA-resistant eGFP-CCDC32(FL) to generate eGFP-CCDC32(Δ78–98). Finally, eGFP, eGFP-CCDC32(1-54), eGFP-CCDC32(Δ78–98), and eGFP-CCDC32(FL) were separately cloned into a pLVx-CMV100 vector (*Dean et al., 2016*) using NEBuilder HiFi DNA Assembly Master Mix (Catalog #E2621). A full list of primers used for mutagenesis and cloning is available in *Table 2*. Note that our disease mimic construct CCDC32(1-54) does not contain a 9 aa peptide (VRGSCLRFQ) in the N-terminus and an extra 12 aa in the C-terminus when CFNDS patient mutation was described (p.(Glu64Glyfs*12)) in *Harel et al., 2020*.

**Table 2.** List of primers.

| Mutagenesis | Forward primer | Reverse primer |
|---|---|---|
| *eGFP-CCDC32(Δ78–98)* | GCAGGATTCAGAAGTGTATGACATGC TTCGAACTCTG | CAGAGTTCGAAGCATGTCATACACTT CTGAATCCTGC |
| *eGFP-CCDC32(FL)_siResistant* | CAGGAAGTGACTTCCAAGGATATGTTACGC ACCCTGGCCCAAGCCAAGAAGGA | TCCTTCTTGGCTTGGGCCAGGGTGCGTAACA TATCCTTGGAAGTCACTTCCTG |
| *eGFP-CCDC32(Δ78–98)_siResistant* | TCCCTTGCAGGATTCAGAAGTGTATGATATGTTACGCA CCCTGGCCCAAGCCAAGAA | TTCTTGGCTTGGGCCAGGGTGCGTAACATAT CATACACTTCTGAATCCTGCAAGGGA |
| Cloning | Forward primer | Reverse primer |
| *backbone : pLVx-CMV100* | CGCCGCTAGCGCTAACTAGTATGGTGAGCAA GGGCGAG | GTCGGGCCCTCTAGACTCGAGTTACTGTTCTG CTGCTGCTG |
| *insert gene : eGFP-CCDC32(FL)* | | |
| *backbone : pLVx-CMV100* | CGCCGCTAGCGCTAACTAGTATGGTGAGCAA GGGCGAG | CGGGCCCTCTAGACTCGAGTTACTTGTACAGCTC GTCCATGC |
| *insert gene : eGFP* | | |
| *backbone : pLVx-CMV100* | CGCCGCTAGCGCTAACTAGTATGGTGAGCAA GGGCGAG | TCGGGCCCTCTAGACTCGAGCTACCTCTGGCCTTC ACCTTC |
| *insert gene : eGFP-CCDC32(1-54)* | | |
| *backbone : pLVx-CMV100* | CGCCGCTAGCGCTAACTAGTATGGTGAGCAA GGGCGAG | GTCGGGCCCTCTAGACTCGAGTTACTGTT CTGCTGCTGCTG |
| *insert gene : eGFP-CCDC32(Δ78–98)* | | |

In addition, mRuby-CLCa in a pLVx-IRES-puro vector was generated in our previous study (*Srinivasan et al., 2018*). AP2-α-AD (aa701-938, mouse) in a pGEX-2T-1 vector and AP2-β-AD (aa592-937, rat) in a pGEX-4T-1 vector were kind gifts of the late Linton Traub (University of Pittsburgh, PA).

## Cell culture, lentivirus infection, siRNA transfection, and rescue

All cell lines used in this study were routinely tested and confirmed negative for mycoplasma contamination. Cell line identities were authenticated by ATCC. None of the cell lines used are listed among commonly misidentified cell lines maintained by the International Cell Line Authentication Committee.

ARPE-19 and ARPE19-HPV16 (herein called ARPE-HPV) cells were obtained from ATCC and cultured in DMEM/F12 (Gibco, Catalog #8122502) with 10% FBS. HEK293T cells were obtained from ATCC and cultured in DMEM (Gibco, Catalog #8122070) with 10% FBS. ARPE-HPV cells that stably express eGFP-CLCa were generated in our previous study (*Chen et al., 2020*). ARPE-HPV cells that stably express a fully functional AP2-α-eGFP, in which eGFP is inserted into the flexible linker of AP2, at aa649, were generated in our previous study (*Mino et al., 2020*), and this AP2-α-eGFP construct has been shown to be fully functional.

Lentiviruses encoding mRuby-CLCa were produced in HEK293T packaging cells following standard transfection protocols (*Kutner et al., 2009*) and harvested for subsequent infections to ARPE-HPV cells to generate ARPE-HPV mRuby-CLCa cells. Lentiviruses encoding eGFP and siRNA-resistant, eGFP-CCDC32(1-54), eGFP-CCDC32(Δ78–98), and eGFP-CCDC32(FL) were produced in HEK293T packaging cells and harvested for subsequent infections to ARPE-HPV mRuby-CLCa cells. Infected cells were FACS sorted for homogenous mRuby and eGFP signals after 3 days and passaged for 2 weeks before experiments.

For siRNA-mediated knockdown of CCDC32, ARPE-HPV cells stably expressing eGFP-CLCa were seeded on 6-well plates (250,000 cells/well) and transfected with 2 rounds of siCCDC32 (Silencer Select Pre-designed siRNA ID#: s228444, targets aa98-102 sequence) through 3 days. Cells treated with siControl (Silencer Select Negative Control #1 siRNA, cat#:4390843) were used as negative control. Transfections of siRNA were mediated with Opti-MEM and Lipofectamine RNAi-MAX (Invitrogen) as detailed in *Chen et al., 2020*.

For CCDC32 rescue experiments, ARPE-HPV cells stably expressing eGFP or siRNA-resistant eGFP-CCDC32(FL) eGFP, eGFP-CCDC32(Δ78–98 or eGFP-CCDC32(1-54)) were transfected with siCCDC32 or siControl as described above. Anti-GFP Monoclonal Antibody (Proteintech, # 66002–1-Ig), Anti-Vinculin Polyclonal Antibody (Proteintech, #26520–1-AP), and anti-C15orf57 Polyclonal antibody (Invitrogen, #PA5-98982) were used in Western Blotting to confirm protein expression level and knockdown efficiency.

## Co-immunoprecipitation (co-IP)

ARPE-HPV cells with stable expression of AP2-α-eGFP, eGFP, eGFP-CCDC32(1-54), eGFP-CCDC32(Δ78–98) or eGFP-CCDC32(FL) were cultured in a 15 cm dish. When confluence reached ~90%, cells were washed 3 x with ice-cold PBS and detached with a cell scratcher. Next, cells were spun down at 500 × *g*, 4 °C for 3 min and then resuspended in 1 ml ice-cold lysis buffer (50 mM Tris, pH 7.5, 150 mM NaCl, mM EDTA, 0.5% Triton X-100, 1×protease inhibitor, 1 mM PMSF). After 30 min rotation in a cold room and occasional vortexing, the lysed cells were spun at 500 ×g, 4 °C, for 3 min to remove the nuclei. Subsequently, 50 µl anti-GFP magnetic beads (Biolinkedin, #L1016) were added to cell lysate that contained 0.5 mg proteins (determined by BCA assay). The reaction was allowed to proceed by rotation at 4 °C for 2 hr and then precipitated with DynaMag. The sediments were washed with lysis buffer, and then resuspended and heat-denatured in 2 x Laemmli buffer (supplemented with 5% β-Mercaptoethanol). The final samples were run into SDS-PAGE gels before: (1) being transferred to membranes for Western Blotting; or (2) being sent to the proteomics core facility for mass spectrometry analysis. Anti-α-Adaptin 1/2 Antibody (C-8) (Santa Cruz Biotechnology, Catalog #sc-17771) was used in WB to detect AP2 enrichments.

## Mass spectrometry analysis

Samples were digested overnight with trypsin (Pierce) following reduction and alkylation with DTT and iodoacetamide (Sigma–Aldrich). Following solid-phase extraction cleanup with an Oasis HLB µelution plate (Waters), the resulting peptides were reconstituted in 10 µl of 2% (v/v) acetonitrile (ACN) and

0.1% trifluoroacetic acid in water. 2 µl of each sample were injected onto an Orbitrap Fusion Lumos mass spectrometer (Thermo Electron) coupled to an Ultimate 3000 RSLC-Nano liquid chromatography systems (Dionex). Samples were injected onto a 75 µm i.d., 75 cm long EasySpray column (Thermo), and eluted with a gradient from 0% to 28% buffer B over 90 min. Buffer A contained 2% (v/v) ACN and 0.1% formic acid in water, and buffer B contained 80% (v/v) ACN, 10% (v/v) trifluoroethanol, and 0.1% formic acid in water. The mass spectrometer operated in positive ion mode with a source voltage of 2.4 kV and an ion transfer tube temperature of 275 °C. MS scans were acquired at 120,000 resolution in the Orbitrap and up to 10 MS/MS spectra were obtained in the Orbitrap for each full spectrum acquired using higher-energy collisional dissociation (HCD) for ions with charges 2–7. Dynamic exclusion was set for 25 s after an ion was selected for fragmentation.

Raw MS data files were analyzed using Proteome Discoverer v2.4 (Thermo), with peptide identification performed using Sequest HT searching against the human reviewed protein database from UniProt. Fragment and precursor tolerances of 10 ppm and 0.6 Da were specified, and three missed cleavages were allowed. Carbamidomethylation of Cys was set as a fixed modification and oxidation of Met was set as a variable modification. The false-discovery rate (FDR) cutoff was 1% for all peptides. To generate the volcano plot, Tubulin Beta 6 was used for normalization.

## Protein purification and GST pull-down assay

GST in a pGEX-6P-1 vector, GST-AP2-α-AD in a pGEX-2T-1 vector, and GST-AP2-β-AD in a pGEX-4T-1 vector were transfected and expressed in BL21(DE3) separately, and then affinity purified using GSTrap HP column (Cytiva). The affinity-purified GST fusion proteins were applied to a HiLoad 16/600 Superdex 200 pg column (Cytiva). Target proteins were collected and concentrated using Amicon Ultra-15 10 K Centrifugal filters (Sigma-Aldrich), and stored in 20 mM HEPES, 150 mM NaCl, 1 mM TCEP, pH 7.4.

In GST pull-down assays, purified GST, GST-AP2-α-AD, and GST-AP2-β-AD were pre-bound to anti-GST beads (Biolinkedin, #L-2004). Subsequently, these beads were separately added into 0.5 mg cell lysate of ARPE-HPV eGFP-CCDC32(FL). The reactions were allowed to proceed by rotation at 4 °C for 2 hr, and then the beads were spun down at 1000 × g, 4 °C for 3 min. The sediments were washed with lysis buffer and then heat-denatured in 2 x Laemmli buffer (supplemented with 5% β-Mercaptoethanol) before running into SDS-PAGE gels, which were followed by western blotting analysis.

## Transferrin receptor (TfnR) uptake assay

Internalization of TfnR was quantified by in-cell ELISA following established protocols described (*Chen et al., 2020*; *Conner and Schmid, 2003*; *Reis et al., 2015*; *Srinivasan et al., 2018*). Briefly, ~15,000 cells were seeded on gelatin-coated 96-well plate overnight. The next day, cells were starved in 37°C-warm PBS$^{4+}$ (1×PBS buffer plus 0.2% BSA, 1 mM CaCl$_2$, 1 mM MgCl$_2$, and 5 mM D-glucose) for 30 min. After starvation, the cell medium was replaced with ice-cold PBS$^{4+}$ containing 5 µg/ml HTR-D65 (anti-TfnR mAb; *Schmid and Smythe, 1991*). In parallel, some cells were kept at 4 °C for the measurement cell surface TfnR (denoted as S) and blank controls (denoted as B), while some were incubated in a 37 °C water bath for the indicated times for the measurement of internalized TfnR (denoted as I). For B and I, acid wash (0.2 M acetic acid and 0.2 M NaCl, pH 2.3) was applied to remove surface-bound HTR-D65. After washing with cold PBS, all cells were fixed with 4% PFA (Electron Microscopy Sciences) diluted in PBS for 30 min at 37 °C. Subsequently, the cells were permeabilized with 0.1% Triton X-100 and blocked with Q-PBS (PBS, 2% BSA, 0.1% lysine, and 0.01% saponin, pH 7.4) for 2 hr. Surface-bound and internalized HTR-D65 were detected with HRP Goat anti-Mouse IgG (H+L) (Bio-Rad) and o-phenylenediamine dihydrochloride (OPD, Sigma-Aldrich). Well-to-well variation of cell numbers was accounted for by BCA assays.

## TIRF and Epi-TIRF microscopy

Cells for imaging were seeded on gelatin-coated glass bottom dishes 35 mm (ibidi, #81218–800) for ~12 hr before data acquisition. Live cell imaging was conducted with a Nikon Eclipse Ti2 inverted microscope that was equipped with: (1) an Apo TIRF/100x1.49 Oil objective; (2) a Prime Back Illuminated sCMOS Camera (Prime BSI, 6.5x6.5 µm pixel size and 95% peak quantum efficiency); (3) a M-TIRF module for epifluorescence (Epi) acquisition; (4) an H-TIRF module for TIRF acquisition, where

penetration depth was fixed to 80 nm; (5) an Okolab Cage Incubator for maintaining 37 °C and 5% $CO_2$.

For time-lapse TIRF imaging, 451 consecutive images were acquired at a frame rate of 1 frame/s for single channel or 0.5 frame/s for dual channels. For time-lapse Epi-TIRF imaging, 451 consecutive Epi and TIRF images were acquired nearly simultaneously at a frame rate of 0.66 frame/s. Perfect Focus System (PFS) was applied during time-lapse imaging.

The acquired data was analyzed using cmeAnalysis (*Aguet et al., 2013*; *Jaqaman et al., 2008*) and DASC (*Wang et al., 2020*). Briefly, cmeAnalysis was used to track the lifetime and fluorescence of clathrin-coated structures, and then DASC was applied to unbiasedly classify *bona fide* CCPs vs. abortive coats. Next, the classified CCP tracks were used to calculate CCP invagination $\Delta z$, as previously described (*Wang et al., 2020*). Tracks that overlap with others or deviate from the properties of a diffraction-limited particle were excluded from the analysis.

## Platinum replica electron microscopy (PREM)

The 'unroofing' technique that mechanically removes the upper cell plasma membrane while preserving the ventral plasma membrane with CCSs and other structures for PREM analyses was performed essentially as described previously (*Yang et al., 2022*). Briefly, ARPE19 cells treated with siCCDC32 or siControl on coverslips were quickly transferred into ice-cold PEM buffer (100 mM PIPES–KOH, pH 6.9, 1 mM $MgCl_2$ and 1 mM EGTA) containing 2 µM unlabeled phalloidin (Sigma, #P2141) and 10 µM taxol (Sigma-Aldrich, #T7402) and unroofed by a brief (1 s) ultrasonic burst from a 1/8-inch microprobe positioned at ~45° angle ~3 mm above the coverslip and operated by Misonix XL2020 Ultrasonic Processor at 17–20% of output power. After sonication, the coverslips were immediately fixed with 2% glutaraldehyde in 0.1 M Na-cacodylate buffer, pH 7.3 for at least 20 min at room temperature.

Sample processing for PREM was performed as described previously (*Svitkina, 2022*). In brief, glutaraldehyde-fixed cells were post-fixed by sequential treatment with 0.1% tannic acid and 0.2% uranyl acetate in water, critical-point dried, coated with platinum and carbon, and transferred onto EM grids for observation.

PREM samples were examined using a JEM 1011 transmission electron microscope (JEOL USA, Peabody, MA) operated at 100 kV. Images were acquired by an ORIUS 832.10 W CCD camera (Gatan, Warrendale, PA) and presented in inverted contrast.

Based on the degree of invagination, the shapes of CCSs were classified into: flat CCSs with no obvious invagination; dome-shaped CCSs that had a hemispherical or less invaginated shape with visible edges of the clathrin lattice; and spherical CCSs that had a round shape with the invisible edges of clathrin lattice in 2D projection images. In most cases, the shapes were obvious in 2D PREM images. In uncertain cases, the degree of CCS invagination was determined using images tilted at ±10–20 degrees. The area of CCSs was measured using ImageJ and used for the calculation of the CCS occupancy on the plasma membrane.

## Acknowledgements

We thank the UTSW proteomics core facility for their help with sample processing and analysis. We thank Justin Bi for PREM analysis. We thank Zhenyang Chen for active participation in protein preparation.

## Additional information

### Funding

| Funder | Grant reference number | Author |
| --- | --- | --- |
| National Natural Science Foundation of China | 32200564 | Zhiming Chen |
| Natural Science Foundation of Hunan Province | 2024JJ2045 | Zhiming Chen |

| Funder | Grant reference number | Author |
|---|---|---|
| National Institutes of Health | R35 GM140832 | Tatyana Svitkina |
| National Institutes of Health | GM73165 | Sandra L Schmid |

The funders had no role in study design, data collection and interpretation, or the decision to submit the work for publication.

## Author contributions

Ziyan Yang, Formal analysis, Validation, Investigation, Writing – original draft, Project administration; Changsong Yang, Formal analysis, Investigation, Methodology, Writing – original draft; Zheng Huang, Formal analysis, Investigation, Visualization; Peiliu Xu, Yueping Li, Lu Han, Linyuan Peng, Formal analysis, Validation, Investigation; Xiangying Wei, Resources, Formal analysis, Writing – review and editing; John E Pak, Investigation, Methodology, Writing – original draft; Tatyana Svitkina, Supervision, Funding acquisition, Writing – original draft, Project administration, Writing – review and editing; Sandra L Schmid, Conceptualization, Funding acquisition, Writing – original draft, Writing – review and editing; Zhiming Chen, Conceptualization, Data curation, Formal analysis, Supervision, Funding acquisition, Investigation, Writing – original draft, Project administration, Writing – review and editing

## Author ORCIDs

Sandra L Schmid (ID) https://orcid.org/0000-0002-1690-7024
Zhiming Chen (ID) https://orcid.org/0000-0002-2423-101X

Reviewer #1 (Public review): https://doi.org/10.7554/eLife.107039.3.sa1
Reviewer #2 (Public review): https://doi.org/10.7554/eLife.107039.3.sa2
Reviewer #3 (Public review): https://doi.org/10.7554/eLife.107039.3.sa3
Author response https://doi.org/10.7554/eLife.107039.3.sa4

# Additional files

## Supplementary files

MDAR checklist

## Data availability

The raw Mass Spec analysis data was uploaded to the MassIVE data repository with accession number MSV000095338. Other data generated or analyzed during this study are included in the manuscript and supporting files. The Matlab-based image processing software (cmeAnalysis and DASC) are deposited at *DanuserLab, 2018*.

The following dataset was generated:

| Author(s) | Year | Dataset title | Dataset URL | Database and Identifier |
|---|---|---|---|---|
| Chen Z | 2025 | GNPS - CCDC32 stabilizes clathrin-coated pits and drives their invagination | https://massive.ucsd.edu/ProteoSAFe/dataset.jsp?accession=MSV000095338 | MassIVE, MSV000095338 |

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

## Appendix 1

**Appendix 1—key resources table**

| Reagent type (species) or resource | Designation | Source or reference | Identifiers | Additional information |
|---|---|---|---|---|
| Strain (*E. coli*) | BL21(DE3) | NEB: C2527H | | Used for recombinant protein expression. |
| Cell line (*Homo sapiens*) | HEK293T | ATCC: CRL-3216 | RRID:CVCL_0063 | Used for lentivirus packaging. |
| Cell line (*Homo sapiens*) | ARPE-19 | ATCC: CRL-2302 | RRID:CVCL_0145 | Adult Retinal Pigment Epithelial cell line. |
| Cell line (*Homo sapiens*) | ARPE-19-HPV16 (ARPE-HPV) | ATCC: CRL-2502 | RRID:CVCL_6338 | ARPE-19 cell line immortalized with HPV16. |
| Cell line (*Homo sapiens*) | ARPE-HPV eGFP-CLCa stable cell line | *Chen et al., 2020* | doi:10.1083/jcb.201908189 | ARPE-HPV cells stably expressing eGFP-CLCa, generated in previous study. |
| Cell line (*Homo sapiens*) | ARPE-HPV AP2-α-eGFP stable cell line | *Mino et al., 2020* | doi:10.1111/tra.12755 | ARPE-HPV cells stably expressing a fully functional AP2-α-eGFP (eGFP inserted at aa649). |
| Cell line (*Homo sapiens*) | ARPE-HPV mRuby-CLCa stable cell line | *Srinivasan et al., 2018* | doi:10.1371/journal.pbio.2005377 | Generated by lentiviral infection of ARPE-HPV cells with mRuby-CLCa (from *Srinivasan et al., 2018*) and FACS sorting. |
| Transfected construct (*Homo sapiens*) | eGFP-CCDC32(FL) siRNA-resistant | This study | This study | Generated in this study and available from the corresponding author upon request. Generated by site-directed mutagenesis on #110505; retains wild-type amino acid sequence (aa98-102) but has altered nucleotide sequence to confer siRNA resistance. See *Table 2* for primers. |
| Transfected construct (*Homo sapiens*) | pLVx-CMV100-eGFP | This study | This study | Generated in this study and available from the corresponding author upon request. |
| Transfected construct (*Homo sapiens*) | eGFP-CCDC32(1-54) (disease mimic) | This study | This study | Generated in this study and available from the corresponding author upon request. Our disease mimic construct. Lacks a 9 aa N-terminal peptide (VRGSCLRFQ) and contains an extra 12 aa C-terminal tag compared to the patient mutation described in *Harel et al., 2020*. |
| Transfected construct (*Homo sapiens*) | eGFP-CCDC32(Δ78–98) | This study | This study | Generated in this study and available from the corresponding author upon request. Generated by deleting amino acids 78–98 from the siRNA-resistant eGFP-CCDC32(FL) construct. See *Table 2* for primers. |
| Recombinant DNA reagent | pLVx-CMV100 | *Dean et al., 2016* | doi:10.1016 /j.bpj.2016.01.029 | Backbone vector for cloning eGFP, eGFP-CCDC32(1-54), eGFP-CCDC32(Δ78–98), and eGFP-CCDC32(FL). |
| Recombinant DNA reagent | pGEX-6P-1-GST | Sigma Aldrich | GE28-9546-48 | Vector for expression of GST tag protein. |
| Transfected construct (*Homo sapiens*) | pLVx-CMV100-eGFP | This study | This study | Generated in this study and available from the corresponding author upon request. |

*Appendix 1 Continued on next page*

*Appendix 1 Continued*

| Reagent type (species) or resource | Designation | Source or reference | Identifiers | Additional information |
|---|---|---|---|---|
| Transfected construct (*Homo sapiens*) | pLVx-CMV100-eGFP-CCDC32(1-54) | This study | This study | Generated in this study and available from the corresponding author upon request. |
| Transfected construct (*Homo sapiens*) | pLVx-CMV100-eGFP-CCDC32(Δ78–98) | This study | This study | Generated in this study and available from the corresponding author upon request. |
| Transfected construct (*Homo sapiens*) | pLVx-CMV100-eGFP-CCDC32(FL) | This study | This study | Generated in this study and available from the corresponding author upon request. |
| Transfected construct (*Homo sapiens*) | pLVx-IRES-puro-mRuby-CLCa | *Srinivasan et al., 2018* | doi:10.1371/journal.pbio.2005377 | Generated in a previous study. |
| Recombinant DNA reagent | pGEX-2T-1-AP2-α-AD (aa701-938, mouse) | | | Kind gift of the late Linton Traub (University of Pittsburgh). GST fusion protein expression vector. |
| Recombinant DNA reagent | pGEX-4T-1-AP2-β-AD (aa592-937, rat) | | | Kind gift of the late Linton Traub (University of Pittsburgh). GST fusion protein expression vector. |
| Antibody (Mouse monoclonal) | Anti-GFP (mouse monoclonal) | Proteintech: Cat# 66002–1-Ig | RRID:AB_11182611 | (1:1000) Used for Western Blotting. |
| Antibody (Rabbit polyclonal) | Anti-Vinculin (rabbit polyclonal) | Proteintech: Cat# 26520–1-AP | RRID:AB_2868558 | (1:40,000) Used for Western Blotting (loading control). |
| Antibody (Rabbit polyclonal) | Anti-C15orf57/CCDC32 (rabbit polyclonal) | Invitrogen / Thermo Fisher Scientific: Cat# PA5-98982 | RRID:AB_2813595 | (1:10,000) Used for Western Blotting to detect endogenous CCDC32 and confirm knockdown. |
| Antibody (Mouse monoclonal) | Anti-α-Adaptin 1/2 (C-8) (mouse monoclonal) | Santa Cruz Biotechnology: Cat# sc-17771 | RRID:AB_2274034 | (1:2000) Used in Western Blotting to detect AP2 enrichment after co-IP. |
| Antibody (Mouse monoclonal) | HTR-D65 (anti-TfnR mAb) (mouse monoclonal) | BioXcell | doi:10.1083/jcb.114.5.869 | (1:200) Primary antibody for labeling surface Transferrin Receptor (TfnR). Used at 5 µg/mL. |
| Antibody (Goat polyclonal) | HRP Goat anti-Mouse IgG (H+L) (goat polyclonal) | Bio-Rad: Cat# 1706516 | RRID:AB_2921252 | (1:5000) Secondary antibody for detection in ELISA. |
| Antibody (Rabbit polyclonal) | Anti-GAPDH Rabbit pAb (rabbit polyclonal) | Abclonal: Cat# AC001 | RRID:AB_2619673 | (1:2000) Used for Western Blotting (loading control). |
| Antibody (Rabbit polyclonal) | HRP Goat Anti-Rabbit IgG (H+L) (rabbit polyclonal) | Abclonal: Cat#AS014 | RRID:AB_2769854 | (1:2000) Secondary antibody for detection in WB. |
| Antibody (Mouse polyclonal) | HRP Goat Anti-Mouse IgG (H+L) (mouse polyclonal) | Abclonal: Cat#AS013 | RRID:AB_2768597 | (1:500) Secondary antibody for detection in WB. |

*Appendix 1 Continued on next page*

*Appendix 1 Continued*

| Reagent type (species) or resource | Designation | Source or reference | Identifiers | Additional information |
|---|---|---|---|---|
| Antibody (Rabbit polyclonal) | AP2B1 (rabbit polyclonal) | Proteintech: Cat#15690–1–AP | RRID:AB_2056351 | (1:5000) Used in Western Blotting to detect AP2 enrichment after co-IP. |
| Antibody (Rabbit monoclonal) | AP2M1 (rabbit monoclonal) | Aifang biological: Cat# AF300772 | | (1:1000) Used in Western Blotting to detect AP2 enrichment after co-IP. |
| Antibody (Rabbit monoclonal) | AP2S1 (rabbit monoclonal) | Aifang biological: Cat# AF302671 | | (1:500) Used in Western Blotting to detect AP2 enrichment after co-IP. |
| Recombinant DNA reagent | pEGFP-C1-CCDC32(FL, human) cDNA | Addgene: Cat# 110505 | | Parental plasmid from which mutants were generated. |
| Sequence-based reagent | siRNA: siCCDC32 | Thermo Fisher Scientific (Silencer Select): ID#: s228444 | | Targets nucleotide sequence encoding aa98-102 of human CCDC32; Used for knockdown. |
| Sequence-based reagent | siRNA: siControl (negative control) | Thermo Fisher Scientific (Silencer Select): Cat#: 4390843 | | Silencer Select Negative Control #1 siRNA. |
| Commercial assay or kit | NEBuilder HiFi DNA Assembly Master Mix | New England Biolabs: Cat# E2621 | | Used for cloning constructs into pLVx-CMV100. |
| Commercial assay or kit | Anti-GFP Magnetic Beads | Biolinkedin: Cat# L1016 | | Used for immunoprecipitation of GFP-tagged proteins. |
| Commercial assay or kit | BCA Protein Assay Kit | Thermo Fisher: Cat# 23225 | | Used to determine protein concentration in lysates. |
| Commercial assay or kit | GSTrap HP Column | Cytiva: Cat# 17528201 | | Used for affinity purification of GST-fusion proteins. |
| Commercial assay or kit | Anti-GST Beads | Biolinkedin: Cat# L-2004 | | Used for GST pull-down assays. |
| Chemical compound, drug | Triton X-100 | BBI: Cat# A600198-0500 | | Used at 0.5% in lysis buffer. |
| Chemical compound, drug | Phalloidin | Sigma-Aldrich: Cat# P2141 | | Used in PEM buffer at 2 µM for cytoskeleton stabilization during unroofing. |
| Chemical compound, drug | Taxol (Paclitaxel) | Sigma-Aldrich: Cat# T7402 | | Used in PEM buffer at 10 µM for microtubule stabilization during unroofing. |
| Software, algorithm | Proteome Discoverer (v2.4) | Thermo Fisher Scientific | RRID:SCR_014477 | Used for raw MS data analysis. |
| Software, algorithm | cmeAnalysis | *Aguet et al., 2013*; *Jaqaman et al., 2008* | GitHub: DanuserLab/ cmeAnalysis | Used for tracking clathrin-coated structures. |
| Software, algorithm | DASC | *Wang et al., 2020* | GitHub: DanuserLab/ cmeAnalysis | Used for unbiased classification of CCPs vs. abortive coats. |

*Appendix 1 Continued on next page*

*Appendix 1 Continued*

| Reagent type (species) or resource | Designation | Source or reference | Identifiers | Additional information |
|---|---|---|---|---|
| Software, algorithm | ImageJ / Fiji | NIH | RRID:SCR_003070 | Used for measuring CCS area and calculating membrane occupancy. |
| Other | Opti-MEM | Thermo Fisher Scientific: Cat# 31985070 | | Reduced-serum medium Used for siRNA transfection. |
| Other | Protease Inhibitor Cocktail | APE ×BIO: Cat# K1015 | | Added to lysis buffer. |
| Other | μ-Dish 35 mm, high Glass Bottom (gelatin-coated) | Ibidi: Cat# 81218–800 | | Used for live-cell imaging. |

