## [Editor Report · eLife Assessment]

The manuscript presents a **valuable** finding that CCDC32, beyond its reported role in AP2 assembly, follows AP2 to the plasma membrane and regulates clathrin-coated pit assembly and dynamics. The authors further identify an alpha-helical region within CCDC32 that is essential for its interaction with AP2 and its cellular function. While live-cell and ultrastructural imaging data are **solid**, future biochemical studies will be needed to confirm the proposed CCDC32-AP2 interaction.

[Editors' note: this paper was reviewed by Review Commons.]

---

## [Referee Report · Reviewer #1 (Public review)]

Yang et al. describes CCDC32 as a new clathrin mediated endocytosis (CME) accessory protein. The authors show that CCDC32 binds directly to AP2 via a small alpha helical region and cells depleted for this protein show defective CME. Finally, the authors show that the CCDC32 nonsense mutations found in patients with cardio-facial-neuro-developmental syndrome (CFNDS) disrupt the interaction of this protein to the AP2 complex. The results presented suggest that CCDC32 may act as both a chaperone (as recently published) and a structural component of the AP2 complex.

---

## [Referee Report · Reviewer #2 (Public review)]

Summary:

The authors responded to my previous concerns with additional arguments and discussion. While I do not object to the publication of this work, two critical experiments are still missing.

Weaknesses:

First, biochemical assays using recombinant proteins should be conducted to determine whether CCDC32 binds to the full AP2 adaptor or to specific AP2 intermediates, such as hemicomplexes. The current co-IP data from mammalian cell lysates are too complex to interpret conclusively. Second, cell fractionation should be performed to assess whether, and how, CCDC32 associates with membrane-bound AP2.

---

## [Referee Report · Reviewer #3 (Public review)]

In this manuscript, Yang et al. characterize the endocytic accessory protein CCDC32, which has implications in cardio-facio-neuro-developmental syndrome (CFNDS). The authors clearly demonstrate that the protein CCDC32 has a role in the early stages of endocytosis, mainly through the interaction with the major endocytic adaptor protein AP2, and they identify regions taking part in this recognition. Through live cell fluorescence imaging and electron microscopy of endocytic pits, the authors characterize the lifetimes of endocytic sites, the formation rate of endocytic sites and pits and the invagination depth, in addition to transferrin receptor (TfnR) uptake experiments. Binding between CCDC32 and CCDC32 mutants to the AP2 alpha appendage domain is assessed by pull down experiments.

Together, these experiments allow deriving a phenotype of CCDC32 knock-down and CCDC32 mutants within endocytosis, which is a very robust system, in which defects are not so easily detected. A mutation of CCDC32, mimicking CFNDS mutations, is also addressed in this study and shown to have endocytic defects.

An experimental proof for the resistance of the different CCDC32 mutants to siRNA treatment would have helped to strengthen the conclusions.

In summary, the authors present a strong combination of techniques, assessing the impact of CCDC32 in clathrin mediated endocytosis and its binding to AP2.

---

## [Author Response]

The following is the authors’ response to the original reviews

**Reviewer #1 (Public review):**
This is a revision of a manuscript previously submitted to Review Commons. The authors have partially addressed my comments, mainly by expanding the introduction and discussion sections. Sandy Schmid, a leading expert on the AP2 adaptor and CME, has been added as a co-corresponding author. The main message of the manuscript remains unchanged. Through overexpression of fluorescently tagged CCDC32, the authors propose that, in addition to its established role in AP2 assembly, CCDC32 also follows AP2 to the plasma membrane and regulates CCP maturation. The manuscript presents some interesting ideas, but there are still concerns regarding data inconsistencies and gaps in the evidence.

With due respect, we would argue that a role for CCDC32 in AP2 assembly is hardly ‘established’. Rather a single publication reporting its role as a co-chaperone for AAGAP appeared while our manuscript was under review. We find some similar and some conflicting results, which are described in our revised manuscript. However, in combination our two papers clearly show that CCDC32, a previously unrecognized endocytic accessory protein, deserves further study.

(1) eGFP-CCDC32 was expressed at 5-10 times higher levels than endogenous CCDC32. This high expression can artificially drive CCDC32 to the cell surface via binding to the alpha appendage domain (AD)-an interaction that may not occur under physiological conditions.

While we acknowledge that overexpression of eGFP-CCDC32 could result in artificially driving it to CCPs, we do not believe this is the case for the following reasons:

i. The bulk of our studies (Figures 2-4) demonstrate the effects of siRNA knockdown on CCDC32 on CCP early stages of CME, and so it is likely that these functions require the presence of endogenous CCDC32 at nascent CCPs as detected with overexpressed eGFP-CCDC32 by TIRF imaging.

ii. At these levels of overexpression eGFP-CCDC32 fully rescues the effects of siRNA KD of endogenous CCCDC32 of Tfn uptake and CCP dynamics (Figure 6F,G). If the protein was artificially recruited to the AP2 appendage domain, one would expect it to compete with the recruitment of other EAPS to CCPs and hence exhibit defects in CCP dynamics. Indeed, we see the opposite: CCPs that are positive for eGFP-CCDC32 show normal dynamics and maturation rates, while CCPs lacking eGFP-CCDC32 are short-lived and more likely to be aborted (Figure 1C).

iii. We have identified two modes of binding of CCDC32 to AP2 adaptors: one is through canonical AP2-AD binding motifs, the second is through an a-helix in CCDC32 that, by modeling, docks only to the open conformation of AP2. Overexpressed CCDC32 lacking this a-helix is not recruited to CCPs (Fig. 6 D,E), indicating that the canonical AP2 binding motifs are not sufficient to recruit CCDC32 to CCPs, even when overexpressed.

(2) Which region of CCDC32 mediates alpha AD binding? Strangely, the only mutant tested in this work, Δ78-98, still binds AP2, but shifts to binding only mu and beta. If the authors claim that CCDC32 is recruited to mature AP2 via the alpha AD, then a mutant deficient in alpha AD binding should not bind AP2 at all. Such a mutant is critical for establish the model proposed in this work.

We understand the reviewer’s confusion and thus devoted a paragraph in the discussion to this issue. As revealed by AlphaFold 3.0 modeling (Figure S6) binding of CCDC32 to the alpha AD likely occurs via the 2 canonical AP2-AD binding motifs encoded in CCDC32. Given the highly divergent nature of AP2-AD binding motifs, we did not identify these motifs without the AlphaFold 3.0 modeling. While these interactions could be detected by GST-pull downs, they are apparently not of sufficient affinity to recruit CCDC32 to CCPs in cells. In the text, we now describe the a-helix we identified as being essential of CCP recruitment as ‘a’ AP2 binding site on CCDC32 rather than ‘the’ AP2 binding site. Interestingly, and also discussed, Alphafold 3.0 identifies a highly predicted docking site on a-adaptin that is only accessible in the open, cargo-bound conformation of intact AP2. This is also consistent with the inability of CCDC32(D78-99) to bind the a:µ2 hemi-complex in cell lysates.

We agree that further structural studies on CCDC32’s interactions with AP2 and its targeting to CCPs will be of interest for future work.

(3) The concept of hemicomplexes is introduced abruptly. What is the evidence that such hemicomplexes exist? If CCDC32 binds to hemicomplexes, this must occur in the cytosol, as only mature AP2 tetramers are recruited to the plasma membrane. The authors state that CCDC32 binds the AD of alpha but not beta, so how can the Δ78-98 mutant bind mu and beta?

We introduced the concept of hemicomplexes based on our unexpected (and now explicitly stated as such) finding that the CCDC32(D78-99) mutant efficiently co-IPs with a b2:µ2 hemicomplex. As stated, the efficiency of this pulldown suggests that the presumed stable AP2 heterotetramer must indeed exist in equilibrium between the two a:s2 and b2:µ2 hemicomplexes, such that CCDC32(D78-99) can sequester and efficiently co-IP with the b2:µ2 hemicomplex. A previous study, now cited, had shown that the b2:µ2 hemicomplex could partially rescue null mutations of a in *C. elegans* (PMID: 23482940). We do not know how CCDC32 binds to the b2:µ2 hemicomplex and we did not detect these interactions using AlphaFold 3.0. However, these interactions could be indirect and involve the AAGAB chaperone. It is also likely, based on the results of Wan et al. (PMID: 39145939), that the binding is through the µ2 subunit rather than b2. As mentioned above, and in our Discussion, further studies are needed to define the complex and multi-faceted nature of CCDC32-AP2 interactions.

(4) The reported ability of CCDC32 to pull down AP2 beta is puzzling. Beta is not found in the CCDC32 interactome in two independent studies using 293 and HCT116 cells (BioPlex). In addition, clathrin is also absent in the interactome of CCDC32, which is difficult to reconcile with a proposed role in CCPs. Can the authors detect CCDC32 binding to clathrin?

Based on the studies of Wan et al. (PMID: 39145939), it is likely that CCDC32 binds to µ2, rather than to the b2 in the b2:µ2 hemicomplex. As to clathrin being absent from the CCDC32 pull down, this is as expected since the interactions of clathrin even with AP2 are weak in solution (as shown in Figure 5C, clathrin is not detected in our AP2 pull down) so as not to have spontaneous assembly of clathrin coats in the cytosol. Rather these interactions are strengthened by both the reduction in dimensionality that occurs on the membrane and by avidity of multivalent interactions. For example, Kirchausen reported that 2 AP2 complexes are required to recruit one clathrin triskelion to the PM.

(5) Figure 5B appears unusual-is this a chimera?

Figure 5B shows an internal insertion of the eGFP tag into an unstructured region in the AP2 hinge. As we have previously shown (PMID: 32657003), this construct, unique among other commonly used AP2 tags, is fully functional. We have rearranged the text in the Figure legend to make this clearer.

Figure 5C likely reflects a mixture of immature and mature AP2 adaptor complexes.

This is possible, but mature heterotetramers are by far the dominant species, otherwise the 4 subunits would not be immuno-precipitated at near stoichiometric levels with the a subunit. Near stoichiometric IP with antibodies to the a-AD have been shown by many others in many cell types.

(6) CCDC32 is reduced by about half in siRNA knockdown. Why not use CRISPR to completely eliminate CCDC32 expression?

Fortuitously, partial knockdown was essential to reveal this second function of CCDC32, as we have emphasized in our Discussion. Wan et al, used CRISPR to knockout CCDC32 and reveal its essential role as a AAGAB co-chaperone. In the complete absence of CCDC32 mature AP2 complexes fail to form. However, under our conditions of partial CCDC32 depletion, the expression of AP2 heterotetramers is unaffected revealing a second function of CCDC32 at early stages of CME. We expect that the co-chaperone function of CCDC32 is catalytic, while its role in CME is more structural; hence the different concentration dependencies, the former being less sensitive to KD than the latter. This is one reason that many researchers are turning to CRISPRi for whole genome perturbation studies as many proteins play multiple roles that can be masked in KO studies.

**Reviewer #2 (Public review):**
Yang et al. describes CCDC32 as a new clathrin mediated endocytosis (CME) accessory protein. The authors show that CCDC32 binds directly to AP2 via a small alpha helical region and cells depleted for this protein show defective CME. Finally, the authors show that the CCDC32 nonsense mutations found in patients with cardio-facial-neuro-developmental syndrome (CFNDS) disrupt the interaction of this protein to the AP2 complex. The results presented suggest that CCDC32 may act as both a chaperone (as recently published) and a structural component of the AP2 complex.Strengths:The conclusions presented are generally well supported by experimental data and the authors carefully point out the differences between their results and the results by Wan et al. (PNAS 2024).Weaknesses:The experiments regarding the role of CCDC32 in CFNDS still require some clarifications to make them clearer to scientists working on this disease. The authors fail to describe that the CCDC32 isoform they use in their studies is different from the one used when CFNDS patient mutations were described. This may create some confusion. Also, the authors did not discuss that the frame-shift mutations in patients may be leading to nonsense mediated decay.

As requested we have more clearly described our construct with regard to the human mutations and added the possibility of NMD in the context of the human mutations.

**Reviewer #3 (Public review):**
In this manuscript, Yang et al. characterize the endocytic accessory protein CCDC32, which has implications in cardio-facio-neuro-developmental syndrome (CFNDS). The authors clearly demonstrate that the protein CCDC32 has a role in the early stages of endocytosis, mainly through the interaction with the major endocytic adaptor protein AP2, and they identify regions taking part in this recognition. Through live cell fluorescence imaging and electron microscopy of endocytic pits, the authors characterize the lifetimes of endocytic sites, the formation rate of endocytic sites and pits and the invagination depth, in addition to transferrin receptor (TfnR) uptake experiments. Binding between CCDC32 and CCDC32 mutants to the AP2 alpha appendage domain is assessed by pull down experiments. While interaction between CCDC32 and the alpha appendage domain of AP2 is clearly described, a discussion of potential association with other AP2 domains would be beneficial to understand the impact of CCDC32 in endocytosis.

The reviewer is correct. That CCDC32 also interacts with other subunits of AP2, is evident from the findings of Wan et al. and by the fact that the CCDC32(D78-99) mutant efficiently co-IPs with the b2:µ2 hemicomplex. We expanded our discussion around this point. CCDC32 remains an, as yet, poorly characterized, but we now believe very interesting EAP worth further study.

Together, these experiments allow deriving a phenotype of CCDC32 knock-down and CCDC32 mutants within endocytosis, which is a very robust system, in which defects are not so easily detected. A mutation of CCDC32, mimicking CFNDS mutations, is also addressed in this study and shown to have endocytic defects.In summary, the authors present a strong combination of techniques, assessing the impact of CCDC32 in clathrin mediated endocytosis and its binding to AP2.
**Recommendations for the authors:**

**Reviewer #2 (Recommendations for the authors):**
(1) The authors must be clear about the differences between the CCDC32 isoform they used in their manuscript and the one used to describe the patient mutations. This could be done, for example, in the methods. This is essential for the capacity of other labs to reproduce, follow up and correctly cite these results.

We have added this information to the Methods.

(2) I believe the authors have misunderstood what nonsense mediated decay is. NMD occurs at the mRNA level and requires a full genome context to occur (introns and exons). The fact that a mutant protein is expressed normally from a construct by no means prove that it does not happen. I believe that adding the possibility of NMD occurring would enrich the discussion.

Thank you, we have now done more homework and have added this possibility into our discussion of the mutant phenotype. However, if a robust NMD mechanism resulted in a complete loss of CCDC42 protein, then the essential co-chaperone function reported by Wan et al, would result in complete loss of AP2. A more detailed characterization of the cellular phenotype of these mutations, including assessing the expression levels of AP2 would be informative.

**Reviewer #3 (Recommendations for the authors):**
- It is not clear what the authors mean by '~30s lifetime cohort' (line 159). They refer to Figure 2H, which shows the % of CCPs. Can the authors explain exactly what kind of tracks they used for this analysis, for example which lifetime variations were accepted? Do they refer to the cohorts in Figure S4? In Figure S4, the most frequent tracks have lifetimes < 20 s (in contrast to what is stated in the main text). Why was this cohort not used?

The ‘30s cohort’ refers to CCPs with lifetimes between 25-35s which encompasses the most abundant species in control cells and CCDC32 KD cells, as shown by the probability curves in Figure 2H. Given the large number of CCPs analyzed we still have large numbers for our analyses n=5998 and 4418, for control and siRNA treated conditions, respectively. Figure 2H shows the frequency of CCPs in cells treated with CCDC32 siRNA are shifted to shorter lifetimes. We have clarified this in the text.

- Figure S1: It is now clear, why the mutant versions of CCDC32 are not detected in this western blot. However, data that show the resistance of these proteins to siCCDC32 is still missing (S1 A is in the absence of siCCSC32 I assume, as the legend suggests). A western blot using an anti-GFP antibody, as the one used in Figure S1, after siRNA knock-known would provide clarity.

That these constructs all contain the same mutation in the siRNA target sequence gives us confidence that they are indeed resistant to siRNA.

- Note that the anti-CCDC32 antibody does not detect the eGFP-CCDC32(∆78-98) as well as full-length and is unable to detect eGFP-CCDC32(1-54)'. This phrase should belong to Figure S1 (B), not (A)

Corrected.

- The immunoprecipitations of CCDC32 and its mutants with AP2 and its subunits are partially confusing. In Figure 5, the authors show that CCDC32 interacts specifically with the alpha-AD, but not with the beta-AD of AP2. In Figure 6B and C, on the other hand, Co-IPs are shown also with the beta and the mu domain of AP2. This is understandable in the context of the full AP2. However, when interaction with the alpha domain (and sigma) is abolished through mutation of helix 78-98, why would beta and mu still interact, when the beta-AD cannot interact with CCDC32 on its own. Are there interaction sites expected outside the ADs in the beta or mu domains?

See responses to reviewer 1 above. This result likely reflects the co-chaperone activity of CCDC32 as reported by Wan et al it likely due to their reported interactions of CCDC32 with the µ2 subnit of b2:µ2 hemicomplexes.

- Figure S6 D, E and F: How much confidence do the authors have on the AlphaFold predictions? Have the same binding poses been obtained repeatedly by independent predictions?

We provide, with a color scale, the confidence score for each interaction, which is very high (>90%). Of course, this is still a prediction that will need to be verified by further structural studies as we have stated.